# Temperature-Induced Critical Phase Transition in Large Language Models

## Abstract

Large Language Models (LLMs) have demonstrated impressive performance. To understand their behaviors, we need to consider the fact that LLMs sometimes show *qualitative changes*. The natural world also presents such changes called *phase transitions*, which are defined by singular, divergent statistical quantities. Therefore, an intriguing question is whether qualitative changes in LLMs are phase transitions. In this work, we have conducted extensive analysis on texts generated by LLMs and suggested that a phase transition occurs in LLMs when varying the temperature parameter. Specifically, statistical quantities have divergent properties just at the point between the low-temperature regime, where LLMs generate sentences with clear repetitive structures, and the high-temperature regime, where generated sentences are often incomprehensible. In addition, *critical* behaviors near the phase transition point, such as a power-law decay of correlation and slow convergence toward the stationary state, are similar to those in natural languages. Our results suggest a meaningful analogy between LLMs and natural phenomena.

## 1 Introduction

Large Language Models (LLMs) have demonstrated impressive performance in various tasks, including machine translation and code generation. Uncovering the origin of the unexpectedly good performance of LLMs is essential both from theoretical and practical perspectives, as it would allow us to devise even better LLMs. A complete theory for LLMs is yet to be developed, but it definitely needs to take into account the fact that LLMs sometimes show *qualitative* changes when varying parameters. Indeed, various qualitative changes play a crucial role in LLMs. For example, it is empirically known that LLMs can exhibit *emergent abilities*: The performance of LLMs sometimes improves unexpectedly when increasing model scales (Brown et al., 2020; Srivastava et al., 2022; Ganguli et al., 2022; Wei et al., 2022), although some argues that the emergent ability may be an artifact due to the arbitrary choice of metrics (Schaeffer et al., 2024). The *grokking* phenomenon (Power et al., 2022), where, during the learning process, the generalization loss drastically decreases long after the convergence of the training loss, is also an example of such qualitative changes that are observed in language models (Liu et al., 2022b; Zhu et al., 2024a).

Qualitative changes are ubiquitous in nature, often expressing themselves as *phase transitions*. For instance, as temperature increases, ferromagnets undergo a phase transition from the ordered phase, in which the magnetic spins are aligned, and the materials are magnetized even with an infinitesimal external magnetic field, to the disordered phase with zero net magnetization. This phase transition accompanies the divergence of the response of spins to the magnetic field, known as susceptibility. Therefore, the phase transition point defined by the divergent susceptibility separates these two phases unambiguously, as illustrated in Fig. 1 (A). This point does not depend on any subjective factors, such as metrics and thresholds.

These raise an intriguing question: Are qualitative changes observed in LLMs phase transitions? If so, it suggests a meaningful analogy between LLMs and natural phenomena. In this study, we numerically investigate the possibility of phase transitions in LLMs. We shall first focus on the change in the statistics of generated texts with the temperature parameter. This parameter is easy to control and known to induce an evident change to generated texts: At low temperatures, clear repetitive structures appear in the texts, whereas, at very high temperatures, they are often incom-

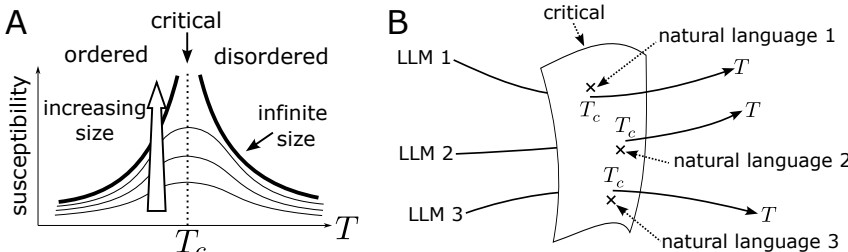

Figure 1: Schematic pictures of phase transitions and critical phenomena in physics and LLMs: (A) Phase transition in a ferromagnetic Ising model, showing how susceptibility exhibits a singularity as a function of temperature in the infinite system size limit. This singular point, which is called the phase transition point, separates the parameter space into ordered and disordered phases. (B) A conjectured relation between LLMs and natural languages within a parameter space where each element represents a distribution of sequences.

prehensible. This behavior reminds us of phase transitions between a periodic, ordered phase and a disordered phase when changing physical temperature. Another motivation is the fact that the decay of correlation in natural languages follows a power law (Li, 1989; Ebeling & Pöschel, 1994; Ebeling & Neiman, 1995; Tanaka-Ishii & Bunde, 2016; Lin & Tegmark, 2017; Takahashi & Tanaka-Ishii, 2017; Shen, 2019; Takahashi & Tanaka-Ishii, 2019; Sainburg et al., 2019; Mikhaylovskiy & Churilov, 2023). Previous studies have measured correlation across different corpora and languages using various metrics, such as the correlation function or the mutual information between words, characters, phones, and others. Despite these variations, the decay in correlation consistently follows a power law, suggesting that this behavior is a universal, necessary condition for texts to be natural. Importantly, the power-law decay of correlation is one of the essential features of *critical phenomena*, which occur near the phase transition point between the ordered and disordered phases. These suggest the existence of a phase transition in LLMs, as illustrated in Fig. 1 (B): The phase transition occurs at a certain temperature $T_c$, at which the LLM exhibits critical behaviors similar to those in natural languages.

To examine the above speculation, we have conducted extensive statistical analysis of sequences generated by LLMs, to detect phase transitions. Our numerical results strongly suggest that a phase transition occurs at temperature $T_c \approx 1$. Below $T_c$, the correlation converges to a positive value, and sequences exhibit repetitive structures. On the other hand, above $T_c$, the correlation converges to zero, and repetitive structures disappear. We further demonstrate that sequences at the transition point have critical properties, such as a power-law decay in a correlation and slow convergence toward the stationary state. Additionally, the observed behaviors exhibit several intriguing features that are not known in typical physical systems: At high temperatures, we find that the correlation does not follow an exponential decay, which is observed in the disordered phases of typical physical systems, and in the low-temperature phase, extremely numerous periodic structures coexist. These findings suggest that LLMs and natural languages are more complex than typical physical systems. Finally, we show natural language datasets have critical properties as well as the LLMs at the transition temperature, implying that the scenario presented in Fig. 1 (B) indeed holds.

The presence of the phase transition in LLMs opens the possibility of understanding LLMs through theories and methods for studying phase transitions. Importantly, our statistical analyses can apply to changes induced by any parameter other than the temperature parameter. We propose a novel, physics-based approach to deepen the theoretical understanding of LLMs.

Our major contributions are the following:

- We propose to formulate qualitative changes in LLMs as phase transitions studied in statistical physics, providing the first strong evidence that practical LLMs exhibit a phase transition.

- We conduct a numerical analysis of the statistical properties of natural language datasets and discuss connections with the criticality of LLMs.

- We show that structures of texts generated at high and low temperatures are peculiar from a statistical-physics perspective, which may be caused by the complex architecture of LLMs with vast numbers of parameters trained on large-scale corpora.

## 1.1 RELATED WORK

**Investigation of the effect of temperature on LLMs:** Previous research has studied how the sampling temperature affects the behavior of LLMs in inference. Renze & Guven (2024) studied the effect of temperature on the performance on problem-solving tasks. Xu et al. (2022); Pursnani et al. (2023); Zhu et al. (2024b), and Grandi et al. (2024) also evaluated the performance on tasks such as code generation, engineering exams, and material selection under different temperature settings. Fradkin et al. (2023) examined how narratives generated by GPT-2 change with temperature from a psychiatric perspective. In contrast to these studies, our work does not focus on the task-specific performance of GPT-2 at various temperatures but rather measures the statistical properties of generated texts to find the existence of phase transitions and critical behaviors.

**Phase transitions in language models:** Emergent abilities and grokking are discussed in connection with phase transitions (Wei et al., 2022; Liu et al., 2022a; Thilak et al., 2022; Nanda et al., 2023; Varma et al., 2023). In these studies, their formulation of phase transitions is not rigorous. Indeed, Schaeffer et al. (2024) has argued that emergent abilities may not be actual qualitative changes. Some theoretical studies formulate emergent abilities, grokking, and other phenomena in language models as phase transitions, yet they consider only mathematical models, not practical LLMs (Žunkovič & Ilievski, 2022; Chang, 2023; Rubin et al., 2024; Cui et al., 2024). Contrary to these earlier studies, we show that a phase transition occurs in practical LLMs. We also note that the earlier studies do not focus on the temperature-induced change in generated texts.

During the preparation of our manuscript, we noticed an independent work by Bahamondes (2023), which measured statistical quantities of sequences of tokens or embedding vectors generated by GPT-2 trained on OpenWebText dataset (Wolfram Research, 2019; Radford et al.). This study claimed that a phase transition occurs at $T \approx 0.1$. However, they did not mention the dependence of correlation on time interval or the asymptotic behavior in the limit of infinite sequence length. We should also note that the phase transition discussed in Bahamondes (2023) would be distinct from the one we study in this work, as the temperature scale is very different.

**Correlation in natural languages and generated texts:** Power-law decays in correlations and mutual information have been observed across different natural language datasets[1] (Li, 1989; Ebeling & Pöschel, 1994; Ebeling & Neiman, 1995; Tanaka-Ishii & Bunde, 2016; Lin & Tegmark, 2017; Takahashi & Tanaka-Ishii, 2017; Shen, 2019; Takahashi & Tanaka-Ishii, 2019; Sainburg et al., 2019; Mikhaylovskiy & Churilov, 2023). These studies, however, did not measure the correlation or mutual information between part-of-speech (POS) tags, which is our focus. Futrell & Levy (2017) and Futrell et al. (2019) measured mutual information between POS tags, yet they did not show the power-law decay. The analysis of correlation and other statistical quantities in texts generated by language models have also been conducted numerically and theoretically in earlier studies (Takahashi & Tanaka-Ishii, 2017; Shen, 2019; Takahashi & Tanaka-Ishii, 2019; Mikhaylovskiy & Churilov, 2023), which concluded that the presence of power-law decays in correlations can depend on models, methods, and setups. These studies have not examined how the correlations depend on parameters such as temperature. Lippi et al. (2019) has studied a power-law decay in a correlation in generated texts with varying temperatures in comparison with natural language datasets. However, the existence of phase transitions was not discussed in their works.

**Criticality in biological systems:** Critical behaviors have also been observed in various biological systems, including neural activity (Munoz, 2018). Previous research has shown that Ising models fitted to neural activity exhibit a phase transition near the temperature where the models are close to the data (Tkacik et al., 2006; Mora & Bialek, 2011). It is noteworthy that their models and data are quite small and simple compared to LLMs.

---

[1]Note that the earlier studies often referred to correlations decaying in a power-law function as *long-range* correlations. In this study, by *long-range*, we mean correlations that converge to a non-zero value.

## 2 PHASE TRANSITIONS AND CRITICAL PHENOMENA

In statistical physics, phase transitions are defined by the singularity of a statistical quantity. To be more precise, a phase transition exists at a parameter point if any of the quantities changes with a singularity at the point, i.e., the quantity or its derivative diverges. We emphasize that a true singularity can exist only in the large-size limit, where the degrees of freedom of the system are infinitely large.

Near the phase transition point, the system can exhibit *critical phenomena*. In this case, the phase transition point is also called the *critical point*. An essential characteristic of critical phenomena is that the correlation decays in a power law. This decay is qualitatively slower than the exponential decay typically observed in ordered or disordered phases. Moreover, because power-law functions are invariant under scale transformations, the power-law decay means the divergence of the length and time scales of the system. This divergence leads to intriguing phenomena: a local perturbation or fluctuation can influence the entire system, the susceptibility or integrated correlation diverges, and the dynamics slow down. Critical behaviors have been discussed in various scientific disciplines, including computer science, biology, finance, and others (Bak et al., 1988; Langton, 1990; Munoz, 2018; Bouchaud, 2024).

In general, theoretically proving the existence of a phase transition is so challenging that it has only been achieved in a few simple theoretical models because we need to consider the large system size limit. However, by estimating statistical quantities with varying system sizes, we can follow the dependence of the quantities on the size to discuss how the quantities diverge. For example, in a ferromagnetic Ising model with a finite number of spins, the susceptibility increases smoothly near the critical temperature. As the system size, i.e., the number of spins, increases, this change becomes sharper and, in the large size limit, it eventually becomes a strictly singular change at the critical point (Domb, 2000), as illustrated in Fig. 1 (A). This transition temperature unambiguously separates the parameter space into two phases, that is, the ordered and the disordered phases. Our approach in this study also follows this method.

## 3 SETUP

We generate texts using pretrained LLMs with temperature sampling (Ackley et al., 1985). Each sequence starts with the single beginning-of-sentence token and ends once the end-of-sentence token appears or when its length reaches the default length limit. The $t$-th token $x_t$ is sampled according to the softmax distribution $P(x_t|x_0, \cdots, x_{t-1}) \propto \exp(-H(x_t|x_0, \cdots, x_{t-1})/T)$ without top-$k$ (Fan et al., 2018) or top-$p$ (Holtzman et al., 2020) sampling strategies. $H(x_t|x_0, \cdots, x_{t-1})$ is the logit of $x_t$, while $T$ is the *temperature* parameter, although it does not necessarily coincide with physical temperature. We then tokenize each of the generated sequences.

To estimate statistical quantities precisely, it is necessary to map each generated text into a sequence of variables that take a small number of states while preserving linguistic information. For this purpose, we map the text to the sequence of universal POS tags. This mapping is useful as it can be easily applied to different languages. Note that our analyses in the following part of this work are applicable to another mapping, such as one-to-semantic tags. In addition, the phase transition is observed with the mapping based on characters, as we will show in App. J. POS tagging transforms the generated texts into POS sequences $(y_0, \cdots, y_{N_{\text{POS}}-1})$, where $y_t$ takes one of 18 different POS tags, that is, 17 universal POS tags plus SPACE. To study the POS sequences of length $N$, we choose POS sequences that are longer than $N$ from all the samples, and analyze the first $N$ tags $(y_0, \cdots, y_{N-1})$ from the beginning for each sequence. For example, a POS sequence of $N_{\text{POS}} = 300$ is used for $N = 256$ but not for $N = 512$. We have confirmed that this sampling method does not qualitatively affect the results presented below, as discussed in App. A. Throughout this paper, we interpret index $t$ as time and a POS sequence $(y_0, \cdots, y_{N-1})$ as a time series. In this case, the length $N$ of the time series corresponds to the system size [2].

---

[2] In POS tagging, subsequent context (interpreted as *future* events) can sometimes influence the POS of preceding words (interpreted as *past* events). Although this may affect our interpretation, the analysis remains valid. Indeed, we have confirmed a similar phase transition is observable when text is mapped to a sequence by replacing each character with a number, as presented in App. J. In this case, the mapping of each character is independent of the mappings of subsequent characters.

Since extensive sampling is needed for precise statistical analysis, we used GPT-2 small with 124M parameters (Radford et al., 2019), using Hugging Face transformers library (Wolf et al., 2020). We note that similar results have been observed for models of 70M, 3161M, and 1B, as presented in Apps. I and J. This implies that larger models also exhibit the phase transition. For POS tagging, we use the spaCy library (Montani et al., 2023) with en_core_web_sm pipeline. We sampled $3.2 \times 10^5$ POS sequences at each temperature. Sampling at any given temperature took about 17 hours using a single NVIDIA A100. The number of samples to compute statistical quantities is $1.6 \times 10^5$, since not all sequences are longer than $N$. Error bars in the figures represent an $80\%$ confidence interval estimated using the symmetric bootstrap-$t$ method (Hall, 1988)[3].

## 4 NUMERICAL RESULTS FOR GPT-2 SEQUENCES

In this section, we present a detailed statistical analysis of sequences generated by GPT-2 to understand the effect of temperature on their structure and dynamics. More specifically, we study the correlation between POS tags, the power spectra of sequences, and the time evolution of POS tag distributions.

### 4.1 CORRELATION BETWEEN TWO TAGS

First, we focus on the time correlation, which is a fundamental statistical quantity that characterizes the structure of a time series. The correlation between two tags, $y_t$ and $y_{t+\Delta t}$ is defined by

$$C_{ab}(t, t + \Delta t) = \mathbb{E}[\delta_{a,y_t} \delta_{b,y_{t+\Delta t}}] - \mathbb{E}[\delta_{a,y_t}] \mathbb{E}[\delta_{b,y_{t+\Delta t}}], \tag{1}$$

where $a$ and $b$ are POS such as NOUN and VERB, $\delta_{a,b}$ is the Kronecker delta, and $\mathbb{E}[\cdot]$ stands for an average over the POS sequences at a fixed temperature $T$. When $y_t$ and $y_{t+\Delta t}$ are independent of each other, $\mathbb{E}[\delta_{a,y_t} \delta_{b,y_{t+\Delta t}}]$ equals $\mathbb{E}[\delta_{a,y_t}] \mathbb{E}[\delta_{b,y_{t+\Delta t}}]$, and this quantity is zero. The dependence of the correlation function on the time interval $\Delta t$ indicates the structure of generated texts. If a text is completely random and disordered, the correlation function decays rapidly with $\Delta t$. In contrast, a text with an order, such as a repetitive structure, exhibits a correlation that remains finite even at large $\Delta t$. Among $18^2$ pairs of $a$ and $b$, we mainly discuss the case with both $a$ and $b$ being the proper noun (PROPN), which has the largest contribution to the phase transition. This is justified by the fact that other pairs with large contributions show similar behaviors (These points are discussed in App. B in detail). Therefore, we simply refer to $C_{\text{PROPN,PROPN}}$ as $C$. Unless otherwise specified, a similar notation is used for other quantities.

Figure 2 shows the correlation $C(t, t + \Delta t)$ at $T = 0.3, 1$, and $1.7$ (see App. C for the results at other temperatures). At $T = 0.3$, the correlation converges to a positive value as the interval $\Delta t$ increases, whereas the plateau value depends on the position $t$ of the former tag. This is typical behavior when a given system has a long-range order. At the boundary temperature of $T = 1$ between low and high temperatures, the correlation decays in a power-law function of $\Delta t$, independent of $t$. This is

---

[3]We attach the codes as supplemental material.

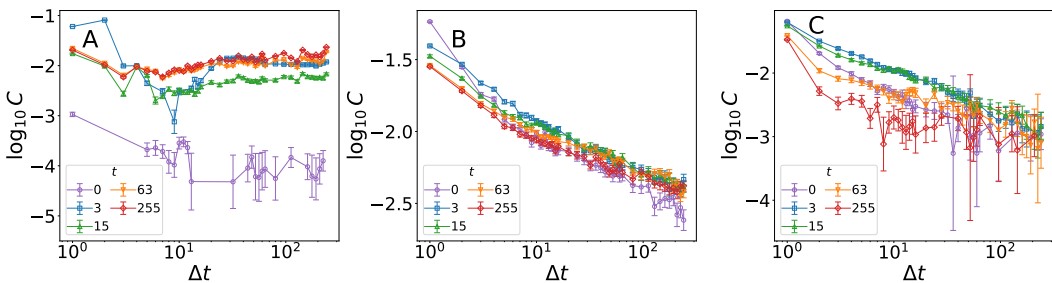

Figure 2: Correlation $C(t, t + \Delta t) = C_{\text{PROPN,PROPN}}(t, t + \Delta t)$ at (A) $T = 0.3$, (B) $T = 1$, and (C) $T = 1.7$ as a function of time interval $\Delta t$, where the sequence length is $N = 512$. Points where the correlation becomes zero have been omitted from the plot to avoid divergences in the logarithmic scale and to focus on significant correlations.

a critical decay with a divergent time scale. In other words, a single word change can significantly influence the overall structure of a text, similar to how ferromagnets behave at the critical point. Apparently, the correlation at $T = 1.7$ also follows a power-law function of $\Delta t$. At first sight, this seems to indicate the system is critical at even high temperatures. However, the correlation also decreases with increasing $t$, meaning that the decay is faster than the critical decay.

To understand the $T$ dependence of the correlation function more systematically, we compute the integrated time correlation:

$$\tau_{ab} = \frac{1}{N} \sum_{t,t'} C_{ab}(t,t') = N \left( \mathbb{E}[m_a m_b] - \mathbb{E}[m_a]\mathbb{E}[m_b] \right), \tag{2}$$

where $m_a = \sum_t \delta_{a,y_t}/N$ is the proportion of POS $a$ in a sequence. When the correlation converges to a finite value or follows a critical decay, $\tau_{ab}$ diverges at $N \to \infty$. Therefore, a finite $\tau_{ab}$ in the $N \to \infty$ limit means that the correlation decays qualitatively faster than a critical decay. We can clearly discriminate between the two distinct behaviors from the $N$ dependence of $\tau_{ab}$.

We show the integrated correlation $\tau = \tau_{\text{PROPN,PROPN}}$ as a function of $T$ in Fig. 3 (A). The integrated correlation increases with decreasing temperature, with a peak at $T \approx 1.1$, and then increases again towards lower temperatures. Figure 3 (B) shows the $N$ dependence more clearly. In the low-temperature regime, $T \lesssim 1$, $\tau$ increases algebraically with $N$, suggesting the divergence in the large size limit $N \to \infty$. On the other hand, at high temperatures $T \gtrsim 1$, it saturates to a finite value. These observations strongly suggest the singular behavior of $\tau$: A critical temperature $T_c \approx 1$ exists, such that in the large size limit $N \to \infty$, the integrated correlation for $T > T_c$ increases as the $T$ approaches $T_c$, eventually diverge at $T = T_c$. This behavior is singular at $T_c$, indicating a *phase transition* at the temperature. The structure of generated texts should be qualitatively different between the low and high temperature regimes[4]. This behavior is similar to that of the susceptibility in ferromagnetic Ising systems in Fig. 1 (A), except that the integrated correlation diverges even below the critical point. Note that if we consider indices $t$ as spatial positions, sequences $(y_0, \cdots, y_{N-1})$ can be regarded as a one-dimensional spin configuration of $N$ spins with nonreciprocal and infinite-range interactions. In this interpretation, the integrated correlation defined by Eq. 2 corresponds to the susceptibility.

In equilibrium statistical mechanics, finite integrated correlations directly indicate asymptotic exponential decay in correlation functions and disordered structures of the system. Meanwhile, the correlation in the POS sequences at high temperatures does not have a simple exponential decay, whereas the integrated correlation $\tau$ converges to a finite value. This means that the behavior of correlation may differ from that observed in the disordered phases of typical physical systems.

---

[4]The peak position of the integrated correlation depends on $N$ and deviates from the exact critical point defined in the $N \to \infty$ limit in general.

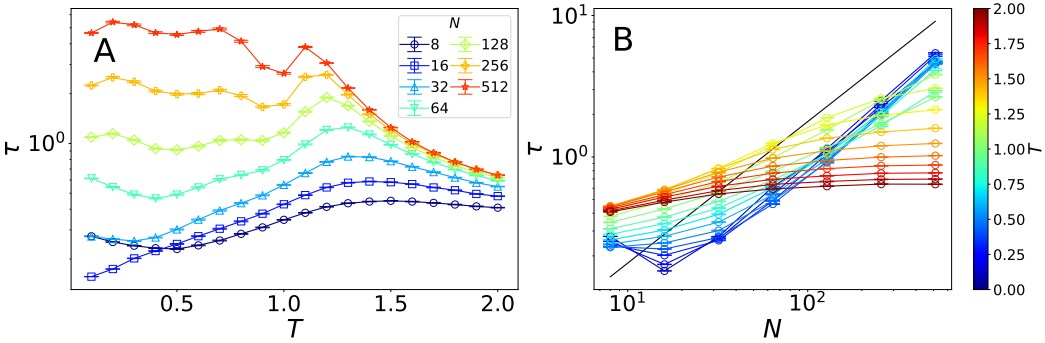

Figure 3: (A) Integrated correlation $\tau = \tau_{\text{PROPN,PROPN}}$ as a function of temperature $T$ for various sequence lengths $N$. (B) The same quantity as a function of sequence length $N$ for various temperatures $T$. The black line represents a line proportional to $N$.

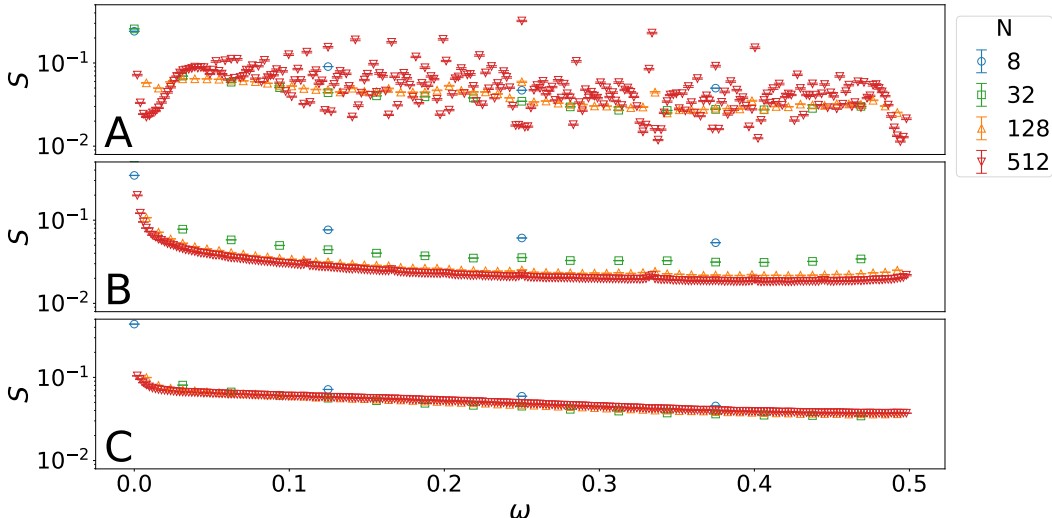

Figure 4: Power spectrum $S = S_{\text{PROPN}}$ of POS sequences as a function of $\omega$ at (A) $T = 0.3$, (B) $T = 1$, and (C) $T = 1.7$. At $T = 0.3$, $S(\omega)$ has many peaks. These peaks disappear at around $T = 1$. At $T = 1.7$, $S(\omega)$ only has a single peak at $\omega = 0$.

## 4.2   POWER SPECTRA OF POS SEQUENCES

Our numerical results raise the question of what structures emerge in the sequences below $T_c$. To answer this question, we compute the power spectrum:

$$S_a(\omega) = N \left( \mathbb{E}\left[ \left| \frac{1}{N} \sum_t e^{-2\pi i \omega t} \delta_{a,y_t} \right|^2 \right] - \left| \mathbb{E}\left[ \frac{1}{N} \sum_t e^{-2\pi i \omega t} \delta_{a,y_t} \right] \right|^2 \right). \tag{3}$$

The power spectrum is the amplitudes of the Fourier modes of the sequences characterized by POS $a$. Therefore, its peak at $\omega$ means that a periodic structure with frequency $\omega$ exists.

The results of $S(\omega) = S_{\text{PROPN}}(\omega)$ at $T = 0.3$, 1, and 1.7 are shown in Fig. 4 (see App. D for other temperatures). The spectrum at high temperatures only has a single peak at $\omega = 0$, suggesting that generated texts lack nontrivial structures. At $T = 1$, near $T_c$, it has multiple small peaks besides the one at $\omega = 0$, but those peaks do not diverge at $N \to \infty$. With further decreasing temperature, $S(\omega)$ has many peaks that grow with $N$, indicating the presence of long-range order. This spectral behavior clearly demonstrates that the well-known repetitive structures emerge only below the critical temperature. However, the analysis reveals a much richer structure than expected: The sequences are superposition of many periodic structures with different frequencies corresponding to distinct peaks. This characteristic power spectrum contrasts with simple periodic structures, which have only a few peaks, observed in the ordered phases of typical physical systems.

## 4.3   TIME EVOLUTION OF POS SEQUENCES

Thus far, we have analyzed the statistical properties of the entire sequences of length $N$ from the beginning. The behavior of correlation $C(t, t + \Delta t)$ in Fig. 2 has suggested that, at small $t$, the distribution of the POS tags strongly depends on $t$, and it gradually approaches its stationary state at large $t$ as the correlation with the initial time decreases. This observation motivates further analysis of the sequences: How the POS tags evolve with time.

We calculate the probability of POS tag at time $t$ being $a$, $v_a(t) = \mathbb{E}[\delta_{y_t,a}]$. Figure 5 shows $v(t) = v_{\text{PROPN}}(t)$ as a function of time at several temperatures. This not only confirms the aforementioned observation but also reveals that the transient time to reach the stationary value greatly varies with $T$. At temperatures far from $T_c$, both higher and lower, $v(t)$ reaches the limiting value rapidly; at low temperatures, it takes a small value, $v \approx 0.1$, while it is a larger one, $v \approx 0.4$, at high temperatures. Near $T_c$, on the other hand, the convergence of $v(t)$ to its stationary state is much

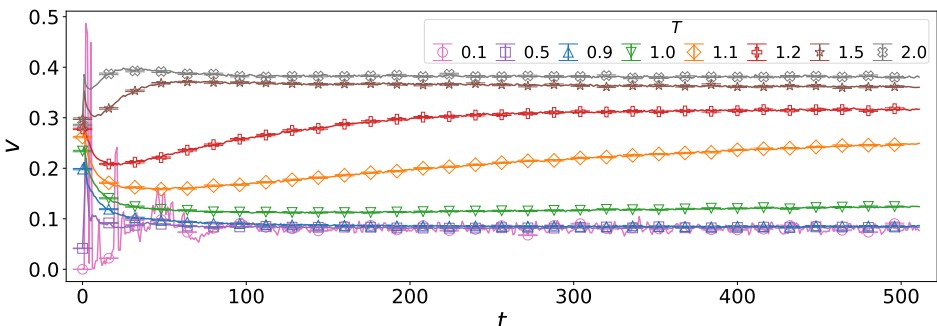

Figure 5: Probability $v(t) = v_{\text{PROPN}}(t)$ that the $t$-th tag is PROPN as a function of time $t$ at $T = 0.1$, 0.5, 0.9, 1, 1.1, 1.2, 1.5, and 2, where the sequence length is $N = 512$. At $T = 0.1$, 0.5, 1.5 and 2, $v(t)$ rapidly reaches the limiting value at $t \lesssim 100$, whereas it needs a much longer time scale to converge when $T = 0.9$, 1.0, 1.1, and 1.2.

slower, indicating the *critical slowing down*, which is the slowing down of dynamics near the critical point. Considering that GPT-2 generates long comprehensible texts around $T_c$, we expect that texts are natural only during the transient time before reaching the stationary state and that texts become too repetitive or incomprehensible once the POS distribution reaches the stationary state. On the other hand, the transient time should diverge when approaching the critical point. If this is indeed the case, then GPT-2 should work well even at temperatures far away from the critical point if the required sequence length is shorter than the transient time. This view is consistent with the empirical fact that LLMs perform well for practical tasks even when the temperature is lower than $T_c \approx 1$ (Renze & Guven, 2024). In general, the divergence of quantities at the critical point follows a power law $\sim |T - T_c|^{-a}$. Accordingly, we estimate the transient time and the exponent $a$ in App. E.

Since $a$ takes 18 different POS tags, $v_a(t)$'s form the 18-dimensional vector $\boldsymbol{v}(t)$. To reveal the underlying dynamics in the entire 18-dimensional space, we employ principal component analysis (PCA). The result shows that $v_{\text{PROPN}}(t)$ dominates the first principal component (PC), meaning that $v_{\text{PROPN}}(t)$ makes an important contribution in the dynamics of $\boldsymbol{v}(t)$. In addition, critical behaviors similar to those of $v_{\text{PROPN}}(t)$ are also observed in the two-dimensional PC space. From these results, we can confirm that it is relevant to focus on $v_{\text{PROPN}}(t)$ (The results are presented in App. F).

## 5 CRITICALITY IN NATURAL LANGUAGE CORPORA

We have shown that generated sequences by GPT-2 have critical properties at $T_c \approx 1$. Empirically, GPT-2 generates comprehensible texts around the same point. From these, it is reasonable to consider that natural languages have statistical properties similar to critical GPT-2. Indeed, earlier studies have found power-law decays in correlations across natural language text and speech corpora (Li, 1989; Ebeling & Pöschel, 1994; Ebeling & Neiman, 1995; Tanaka-Ishii & Bunde, 2016; Lin & Tegmark, 2017; Takahashi & Tanaka-Ishii, 2017; Shen, 2019; Takahashi & Tanaka-Ishii, 2019; Sainburg et al., 2019; Mikhaylovskiy & Churilov, 2023).

To test this idea, we first calculate the perplexity of GPT-2 at various $T$ on OpenWebTextCorpus (OWTC) (Gokaslan & Cohen, 2019)[5]. Figure 6 (A) shows that the perplexity becomes minimum at around $T = 1$, meaning that GPT-2 is the closest to the corpus at the point. And then, we compare the statistical properties of the OWTC with those of the GPT-2 sequences. The correlation function and the integrated correlation of the OWTC, shown in Figs. 6 (B) and (C), respectively, exhibit qualitatively similar behaviors with the GPT-2 sequences near $T_c$; a power-law decay of the correlation and the integrated correlation growing with $N$. As Fig. 6 (D) presents, the time dependence of $v(t)$ for the OWTC is also very similar to that of GPT-2 at $T = 0.9$. Strikingly, both the OWTC and GPT-2 at $T = 0.9$ follow almost the same trajectory in this plot over long time, indicating quantitative agreement in their POS distributions. The power spectrum and the dynamics

---

[5]The perplexity was calculated following the procedure mentioned in the documentation of Transformers (https://huggingface.co/docs/transformers/v4.40.2/en/perplexity).

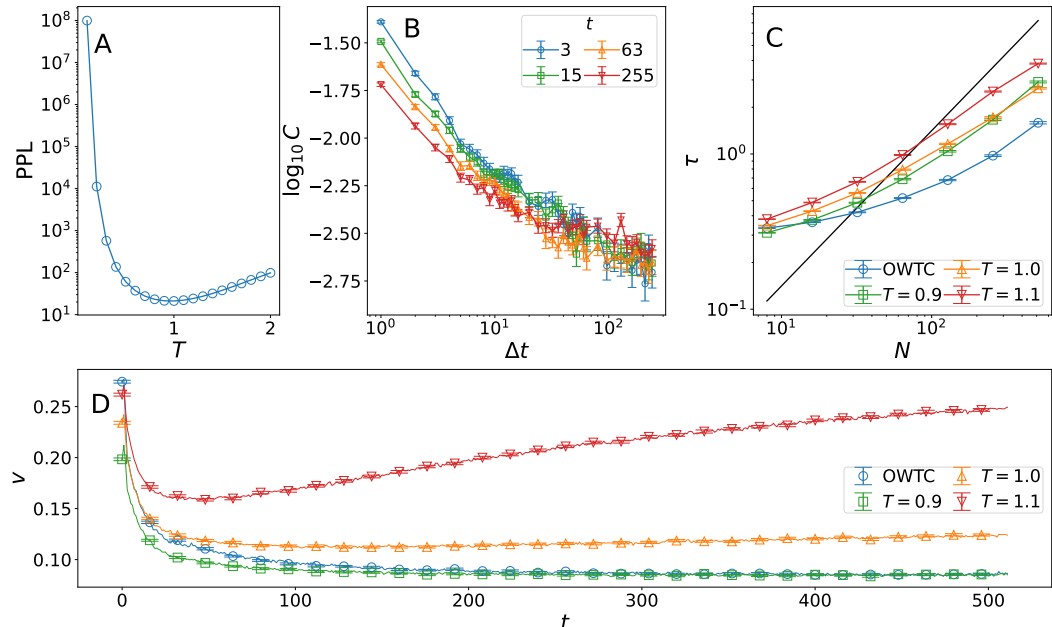

Figure 6: (A) Perplexity of GPT-2 on $10^4$ rows extracted from OWTC with varying temperature $T$. (B) Correlation $C(t, t + \Delta t) = C_{\text{PROPN,PROPN}}(t, t + \Delta t)$ in OWTC as a function of time interval $\Delta t$, where the sequence length is $N = 512$. (C) Integrated correlation $\tau = \tau_{\text{PROPN,PROPN}}$ in OWTC and sequences generated by GPT-2 at $T = 0.9$, $1$, and $1.1$, as a function of sequence length $N$. The black line represents a line proportional to $N$. (D) Probability $v(t) = v_{\text{PROPN}}(t)$ that the $t$-th tag is PROPN in OWTC and sequences generated by GPT-2 at $T = 0.9$, $1$, and $1.1$, as a function of time $t$, where the sequence length is $N = 512$. The results in (B), (C), and (D) were calculated based on $1.6 \times 10^5$ POS sequences.

in the two-dimensional PC space also have similar properties (see App. G for the results). Note that the WikiText dataset (Merity et al., 2016) displays statistical properties similar to those of critical GPT-2 as well, as shown in App. H. We thus expect that the criticality observed here is common across different natural language datasets.

# 6 DISCUSSION

In summary, we have studied the effect of the temperature parameter on the GPT-2 sequences and shown that a critical phase transition occurs at $T_c \approx 1$. We have further demonstrated that the criticality in some statistical quantities appears in natural language corpora as well. It is reasonable that LLMs exhibit critical behaviors similar to those of natural languages near the default temperature $T = 1$, at which logits are not rescaled. However, the existence of the phase transition is nontrivial. Moreover, the high and low temperature regimes far from $T = 1$ show the unexpected behaviors, implying the complex nature of LLMs. Our results also provide a possible picture that, near $T_c$, comprehensible sentences emerge only during the transient time before reaching the stationary state. We now discuss some implications of our numerical results.

**Phase transition in the large context window limit:** Since we have analyzed the statistical properties of sequences shorter than the context window of the LLM, the phenomena discussed here should be observed within its scale. Therefore, to increase the system size $N$, the context window must be longer than $N$. The true phase transition in the $N \to \infty$ limit should exist in the limit where the size of models and the length of texts in the training data are infinitely large.

**Robustness of the phase transition:** The phase transition possibly occurs with varying other parameters because in crossing a phase boundary in a multi-dimensional parameter space, a phase transition is observed, irrespective of the direction. We also expect that the phase transition exists

with other mappings of texts to sequences, other languages, and other LLMs from the following facts: the power-law decay of correlation is universally observed in different mappings and languages, as mentioned in Sec. 1; the behavior of generated texts with varying temperature is similar with different LLMs, that is, generated texts have repetitions at low temperatures while they are incomprehensible at high temperatures; the effect of details of systems is weak near the critical point due to the divergence of scale. Indeed, in Apps. I and J, we conduct the same analyses with a mapping based on characters, and for Japanese GPT-2 medium (361M) (Zhao & Sawada; Sawada et al., 2024) and Pythia (70M and 1B) (Biderman et al., 2023), to obtain similar results. Specifically, the integrated correlation saturates above $T_c \approx 1$ and diverges below it, suggesting the presence of a phase transition. In particular, observing phase transitions in models of different sizes (70M, 124M, 361M, and 1B) suggests that similar phase transitions may also occur in larger models.

**Universality class of LLMs and natural languages:** We have discussed the critical behaviors in sequences generated by GPT-2 at around $T_c$ and shown that GPT-2 and the natural language corpora have common critical behaviors. In the literature of statistical physics, critical phase transitions are classified into *universality classes*, each of which has a distinct set of exponents that describe the behavior of various quantities near the critical point, such as the exponent characterizing the transient time (This is roughly estimated in App. E). If two systems belong to the same universality class, it means they share fundamental characteristics. An interesting direction for future work is to precisely estimate the exponents for different LLMs to identify their universality classes and investigate the relationship between these classes and performance.

**Unique statistical properties at high and low temperatures**: The statistical behavior at high temperatures has a characteristic behavior that is not common in conventional equilibrium disordered phases: Whereas the integrated correlation is convergent, the decay of correlation is not simply exponential. This is due to the correlation decreasing with both $\Delta t$ and $t$. Precisely following the $t$ dependence of $C(t, t+\Delta t)$ with fixed $\Delta t$ is needed to address this point. Whereas we have not fully examined the potential effect of POS tagging at very high temperatures, where generated texts contain many incomprehensible words yet the tagger automatically assigns POS tags, any such effect should be insignificant at low temperatures and near the critical point, which are our main focus.

The low-temperature phase also exhibits an intriguing phenomenon: This phase is definitely ordered in the sense that the correlation converges to a nonzero value. However, unlike conventional ordered phases in equilibrium statistical mechanics, the dynamics of the low-temperature GPT-2 is complex: Sequences have repetitive structures with many peaks in the power spectrum. Up to length $N = 512$, the number of peaks grows with $N$, suggesting that the power spectrum would have continuous support of diverging components. If this observation is confirmed, it implies that the low-temperature sequences are *chaotic* in nature. We have also noticed that, at $T = 0$, GPT-2 performs top-1 sampling and is deterministic, and the correlation must be strictly zero. Therefore, another phase transition could exist at very low temperature between the repetitive and deterministic phases. We cannot identify this temperature scale because fluctuations are too small to precisely follow the $N$ dependence at this temperature regime. Even more extensive calculations are necessary to study this possible transition. We cannot conclude on the connection between this potential transition and the one argued in Bahamondes (2023).

These peculiar phenomena should originate from the distinct features in LLMs: nonreciprocal and infinite-range interactions implemented with attention mechanisms and highly non-uniform parameters trained on large-scale corpora. Importantly, although mathematical models of machine learning have been extensively studied in statistical physics (Carleo et al., 2019), existing models do not have critical phase transitions with the unique properties observed in this study. An important and promising future direction is to explore the development of mathematical models that exhibit phase transitions similar to those observed in this study.

## 7 LIMITATIONS

- The size of the models we analyzed is moderate (70M, 124M, 361M, and 1B). As a future direction, statistical analysis of larger models is important, because the true phase transition seems to exist in the large context window limit as we have discussed.

- Further investigation is needed to better understand how the criticality relates to linguistic properties and practical tasks.

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

## A EFFECT OF SHORT SEQUENCES

In our analysis, short sequences are used only when they are longer than $N$. This does not affect our conclusion, as we mentioned in Sec. 3. Figure 7 shows the correlation between the 0-th tag and the $\Delta t$-th tag for different $N$. The dependence of correlation on $N$ is very small. We also calculate the integrated correlation in a slightly different setting, where only sequences longer than $512$ are used irrespective of $N$. The results, shown in Fig. 8, are similar to those in our original setting. Therefore, we can conclude that the effect of short sequences is not significant.

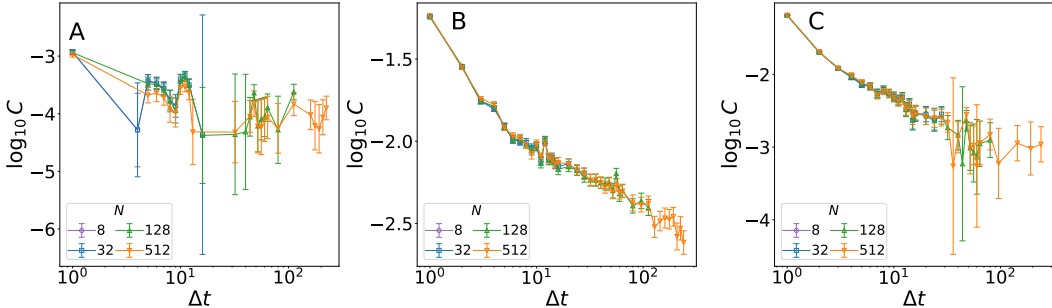

Figure 7: Correlation $C(0, \Delta t) = C_{\text{PROPN,PROPN}}(0, \Delta t)$ for various sequence lengths $N = 8$, $32$, $128$, and $512$, at (A) $T = 0.3$, (B) $T = 1$, and (C) $T = 1.7$, as a function of time interval $\Delta t$.

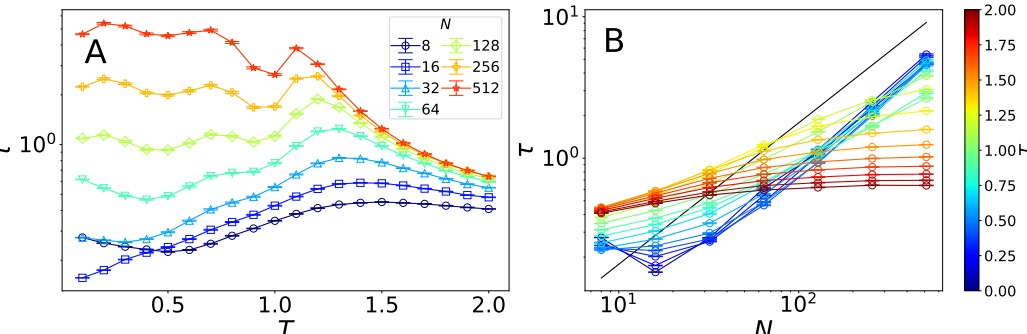

Figure 8: (A) Integrated correlation $\tau = \tau_{\text{PROPN,PROPN}}$ as a function of temperature $T$ for various sequence length $N$. (B) The same quantity as a function of sequence length $N$ for various temperatures $T$. The black line represents a line proportional to $N$. The setup and method for computation are the same as in the main part, except that only sequences longer than $512$ are used irrespective of $N$.

## B INTEGRATED CORRELATIONS WITH THE LARGEST CONTRIBUTIONS

In Sec. 4.1, we noted that correlation $C_{\mathrm{PROPN,PROPN}}$ has the largest contribution to the critical phenomena among correlations $C_{ab}$. To demonstrate it, we show the 10 largest absolute values of integrated correlation $\tau_{ab}$ for each $T$ as a function of $N$ in Fig. 9 and 10. Clearly, the contribution of $\tau_{\mathrm{PROPN,PROPN}}$ is the largest for all temperatures.

To capture the behavior of all integrated correlations roughly, we introduce the square root of the sum of squared integrated correlations, $\bar{\tau} = \sqrt{\sum_{a,b} \tau_{ab}^2}$. Note that the simple sum is trivially zero, i.e., $\sum_{ab} \tau_{ab} = 0$. As shown in Fig. 11, the temperature dependence of $\bar{\tau}$ is similar to that of $\tau_{\mathrm{PROPN,PROPN}}$.

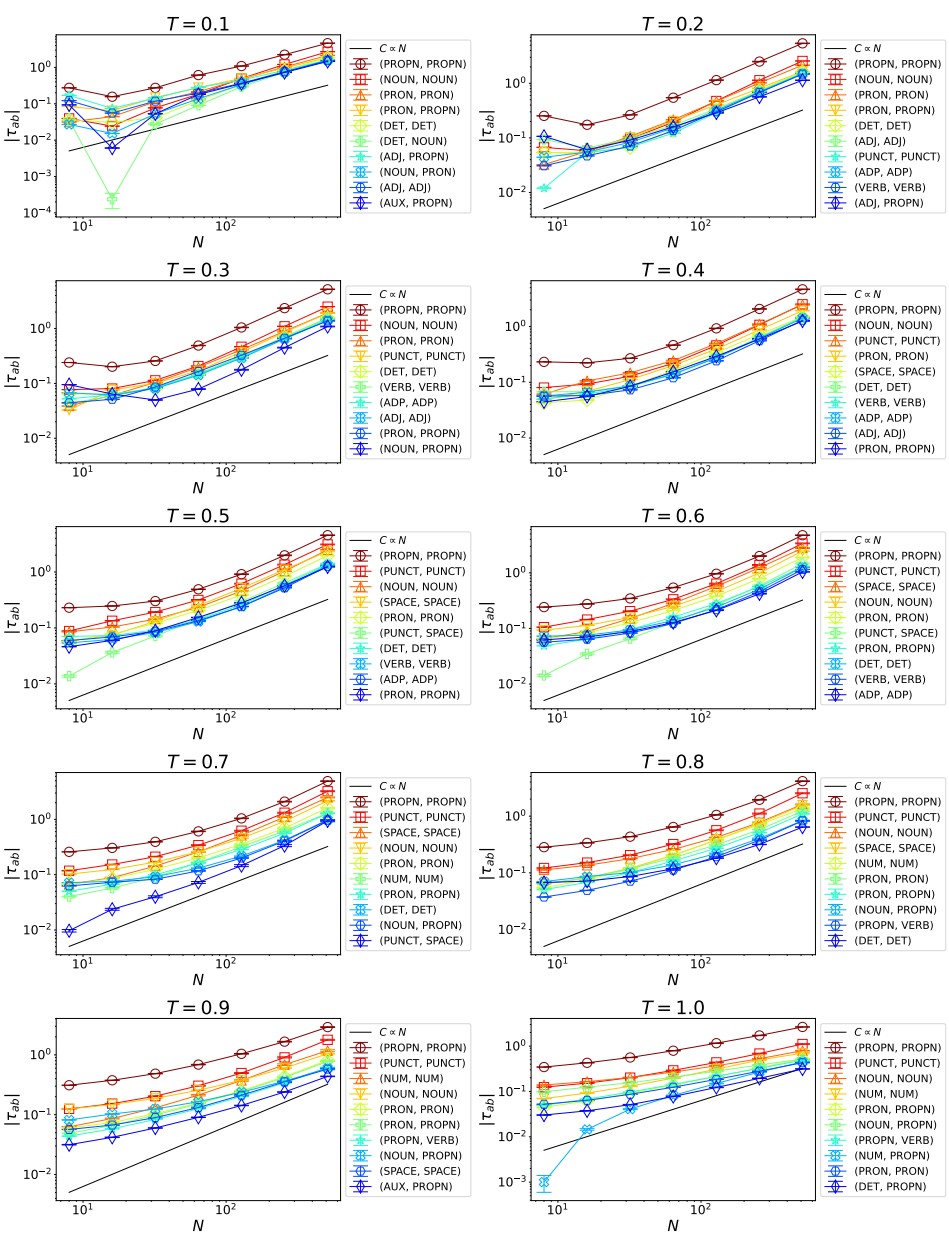

Figure 9: 10 largest absolute values of integrated correlations $\tau_{ab}$ at $T = 0.1, 0.2, \cdots, 1$, as a function of sequence length $N$.

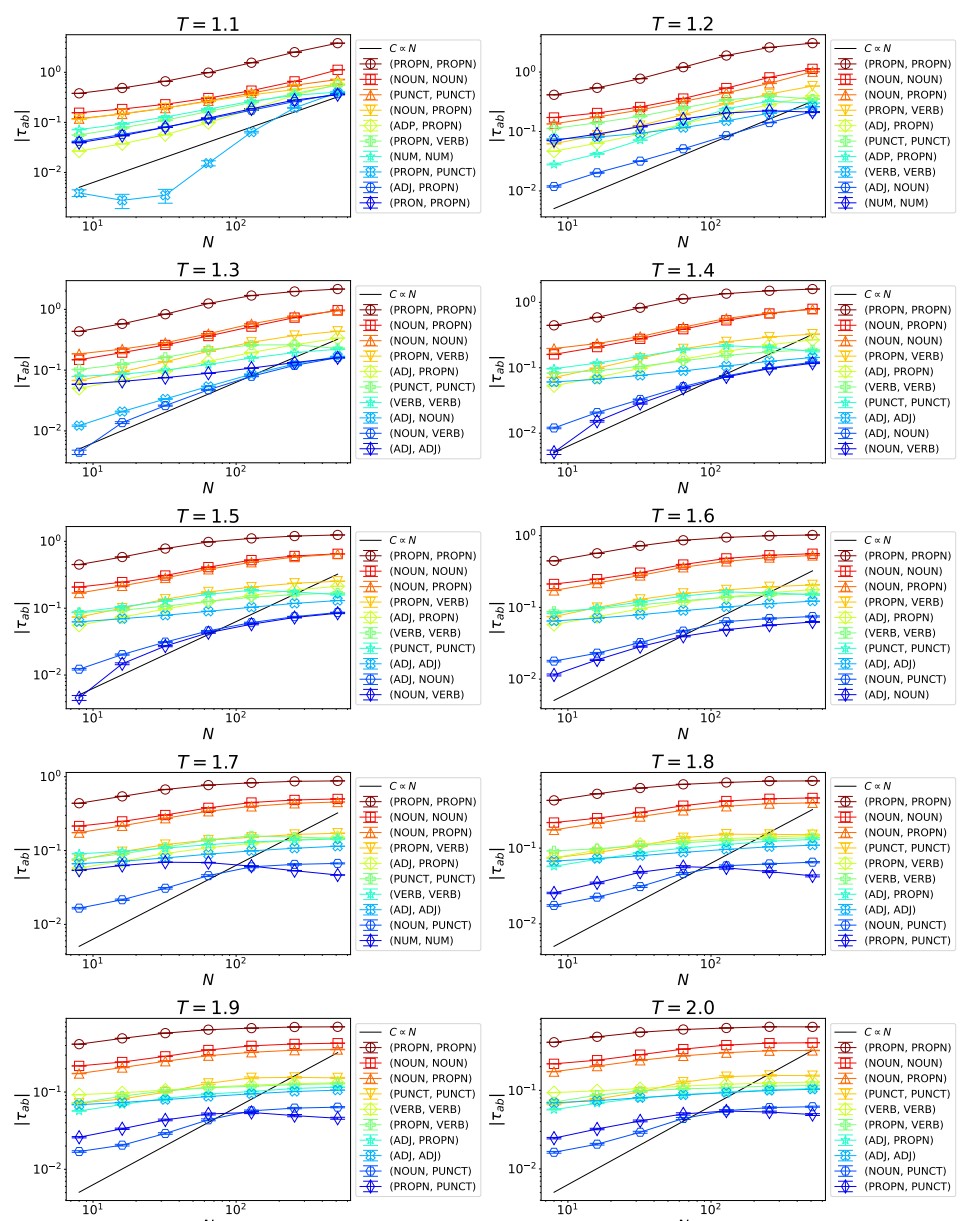

Figure 10: 10 largest absolute values of integrated correlation $\tau_{ab}$ at $T = 1.1, 1.2, \cdots, 2$, as a function of sequence length $N$.

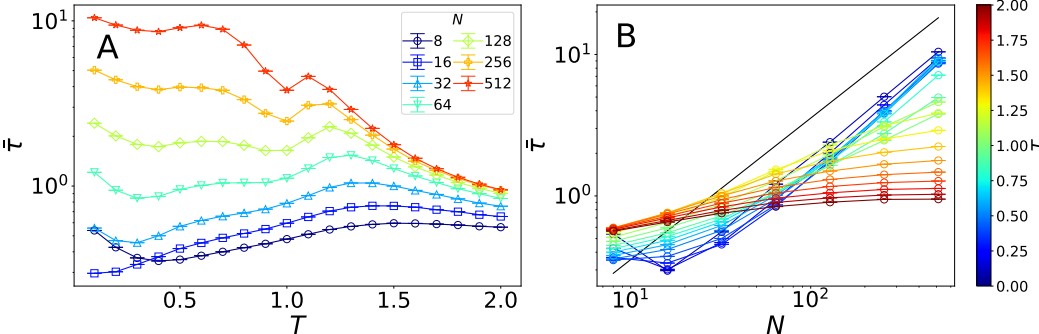

Figure 11: Square root of of sum of squared integrated correlation, $\bar{\tau}$, (A) as a function of $T$ for various $N$ and (B) as a function of $N$ for various $T$. The black line represents a line proportional to $N$.

## C CORRELATIONS

In Sec. 4.1, we have discussed the correlation. However, only results at $T = 0.3$, 1, and 1.7 were presented in that section. In this appendix, we show the correlations $C = C_{\text{PROPN,PROPN}}$ at $T = 0.1, 0.2, \cdots, 2$, see Figs. 12 and 13. The data supports the discussion we made in the main text. The correlation converges to a positive value and to zero at low and high temperatures, respectively. At $T_c \approx 1$ between them, it follows a critical decay, which is a power-law decay with a constant prefactor.

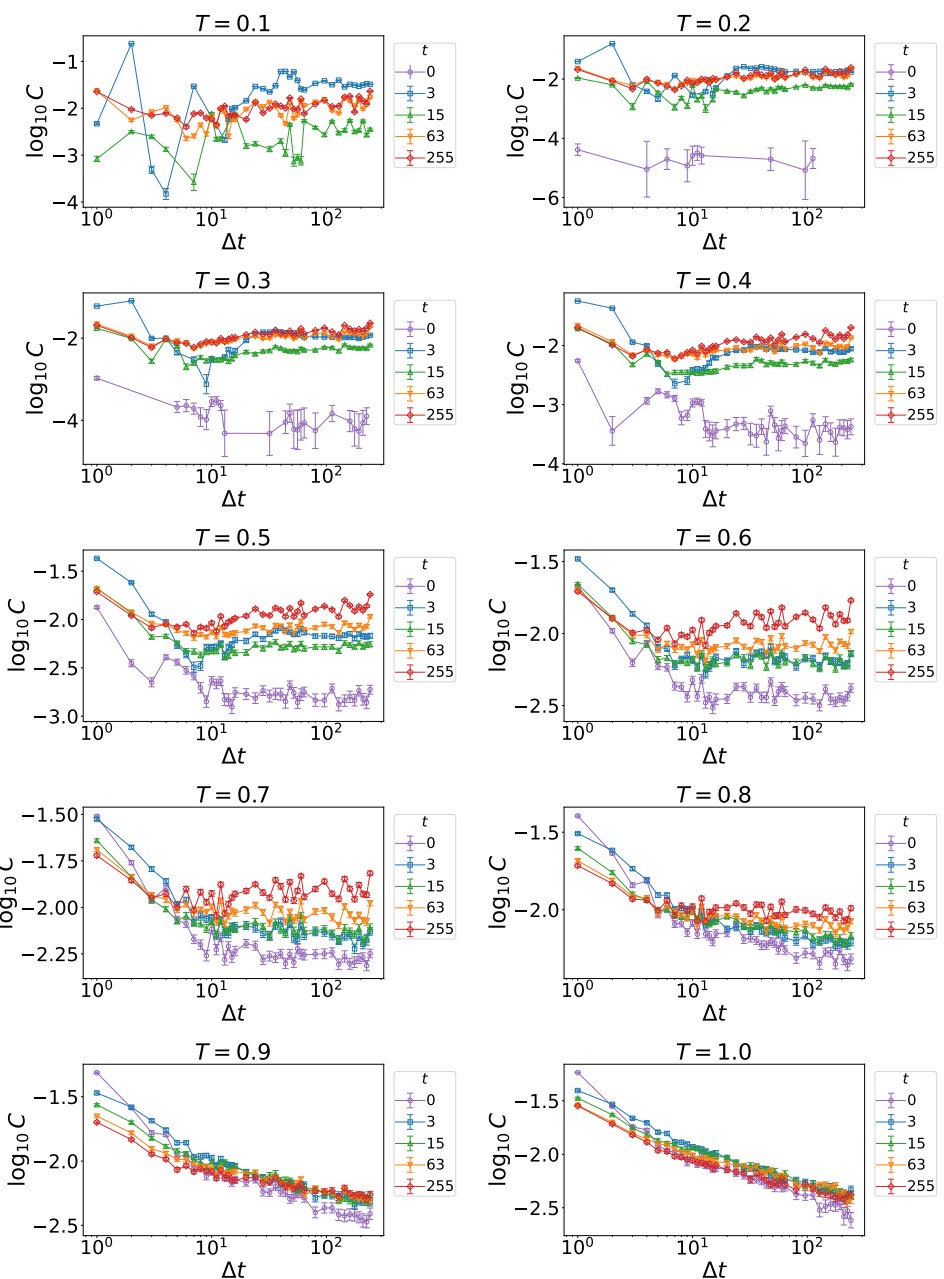

Figure 12: Correlation $C(t, t + \Delta t) = C_{\text{PROPN,PROPN}}(t, t + \Delta t)$ at $T = 0.1, 0.2, \cdots, 1$, as a function of time interval $\Delta t$, where the sequence length is $N = 512$. Points where the correlation becomes zero have been omitted from the plot to avoid divergences in the logarithmic scale and to focus on significant correlations.

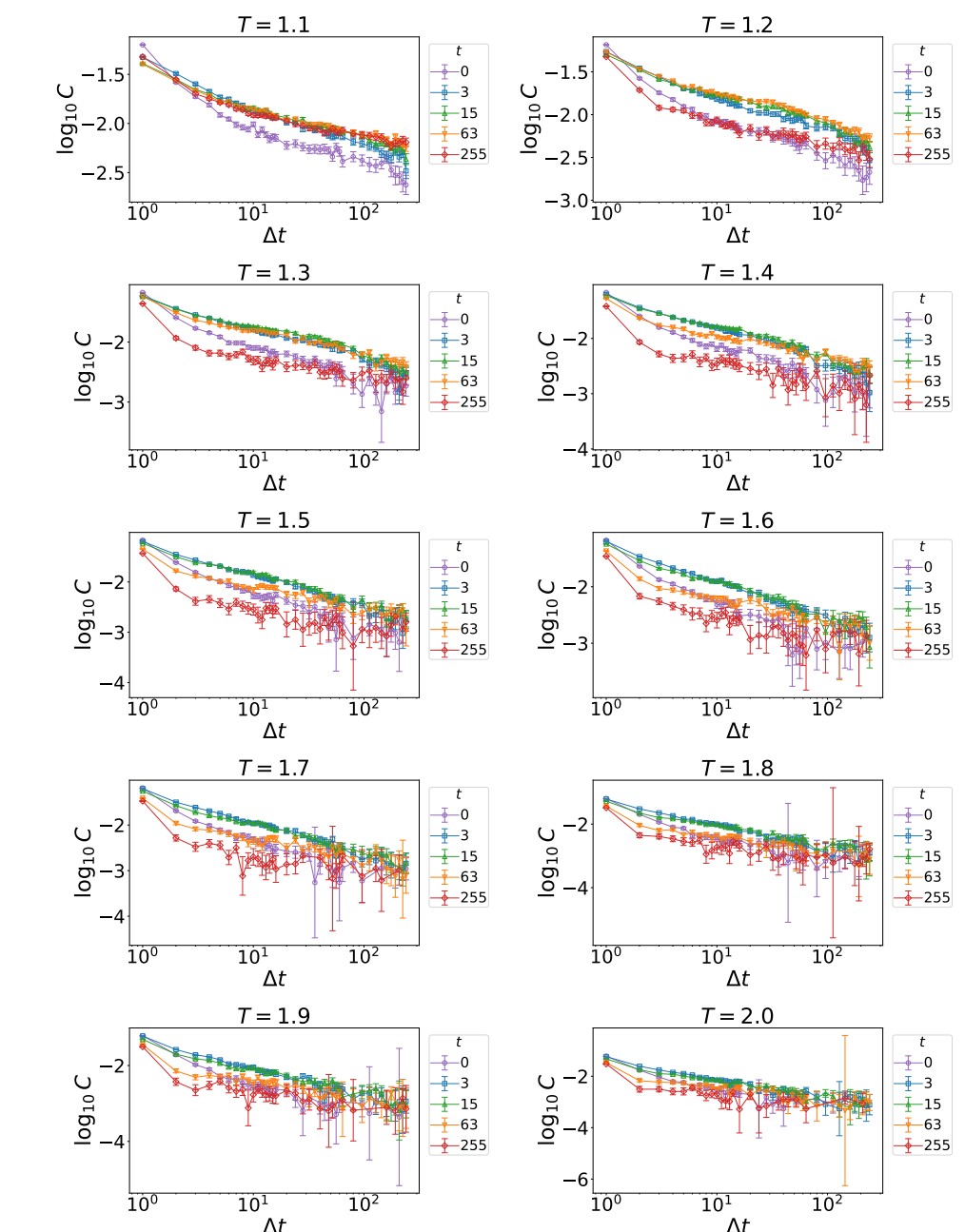

Figure 13: Correlation $C(t, t + \Delta t) = C_{\text{PROPN,PROPN}}(t, t + \Delta t)$ at $T = 1.1, 1.2, \cdots, 2$, as a function of time interval $\Delta t$, where the sequence length is $N = 512$. Points where the correlation becomes zero have been omitted from the plot to avoid divergences in the logarithmic scale and to focus on significant correlations.

## D Power Spectra

In Sec. 4.2, we have discussed the power spectra at $T = 0.3$, 1, and 1.7, and have demonstrated that many peaks growing with $N$ in the power spectra emerge only below the critical point $T_c \approx 1$. Here, we show the power spectra $S$ at $T = 0.1, 0.2, \cdots, 2$ in Figures. 14, 15, 16, and 17. These results support our discussions that the structures of sequences are qualitatively different between the high and low-temperature regimes: At any temperature above $T_c$, the power spectrum has multiple peaks, whereas it has only one peak at $\omega = 0$ at higher temperatures.

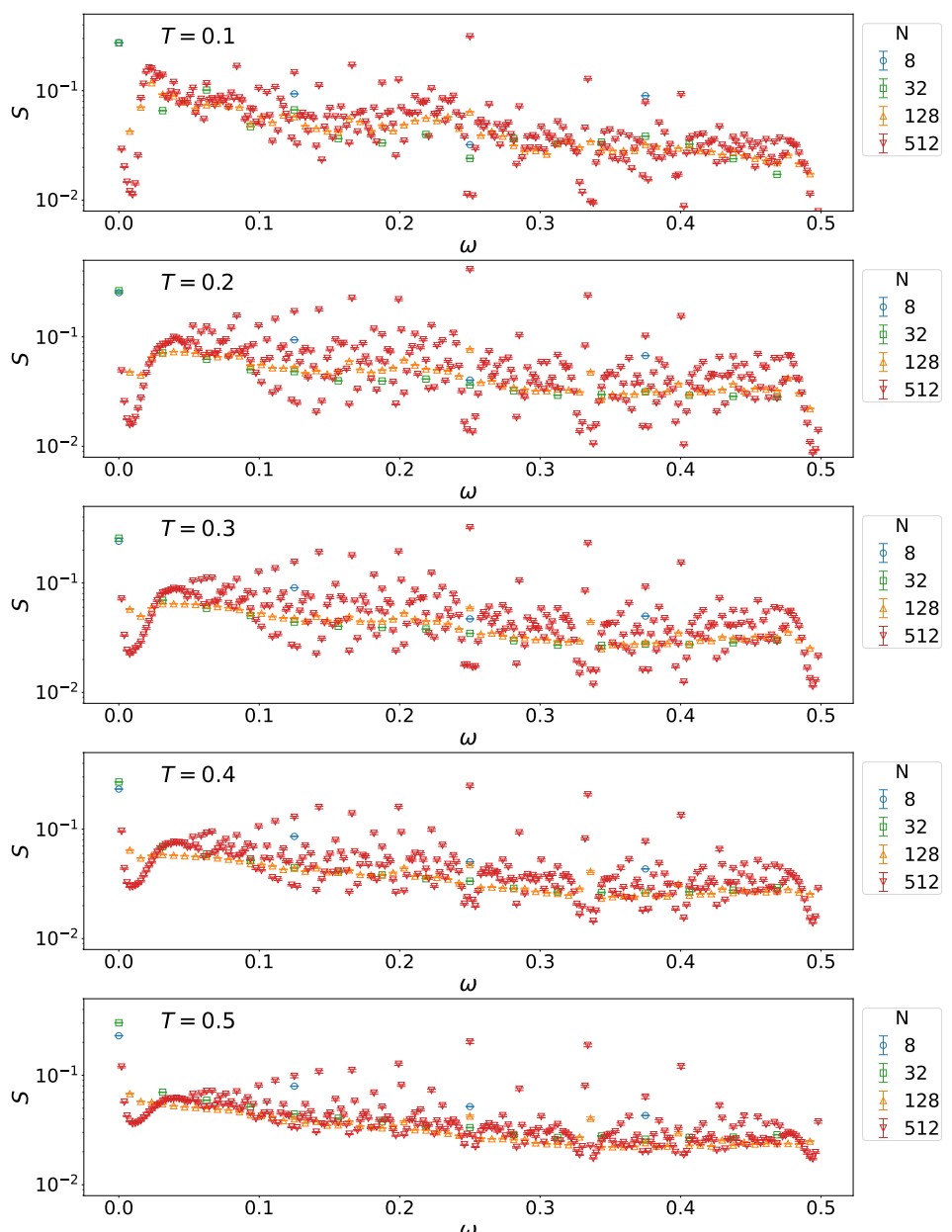

Figure 14: Power spectrum $S = S_{\text{PROPN}}$ as a function of $\omega$ at $T = 0.1, 0.2, 0.3, 0.4$, and $0.5$.

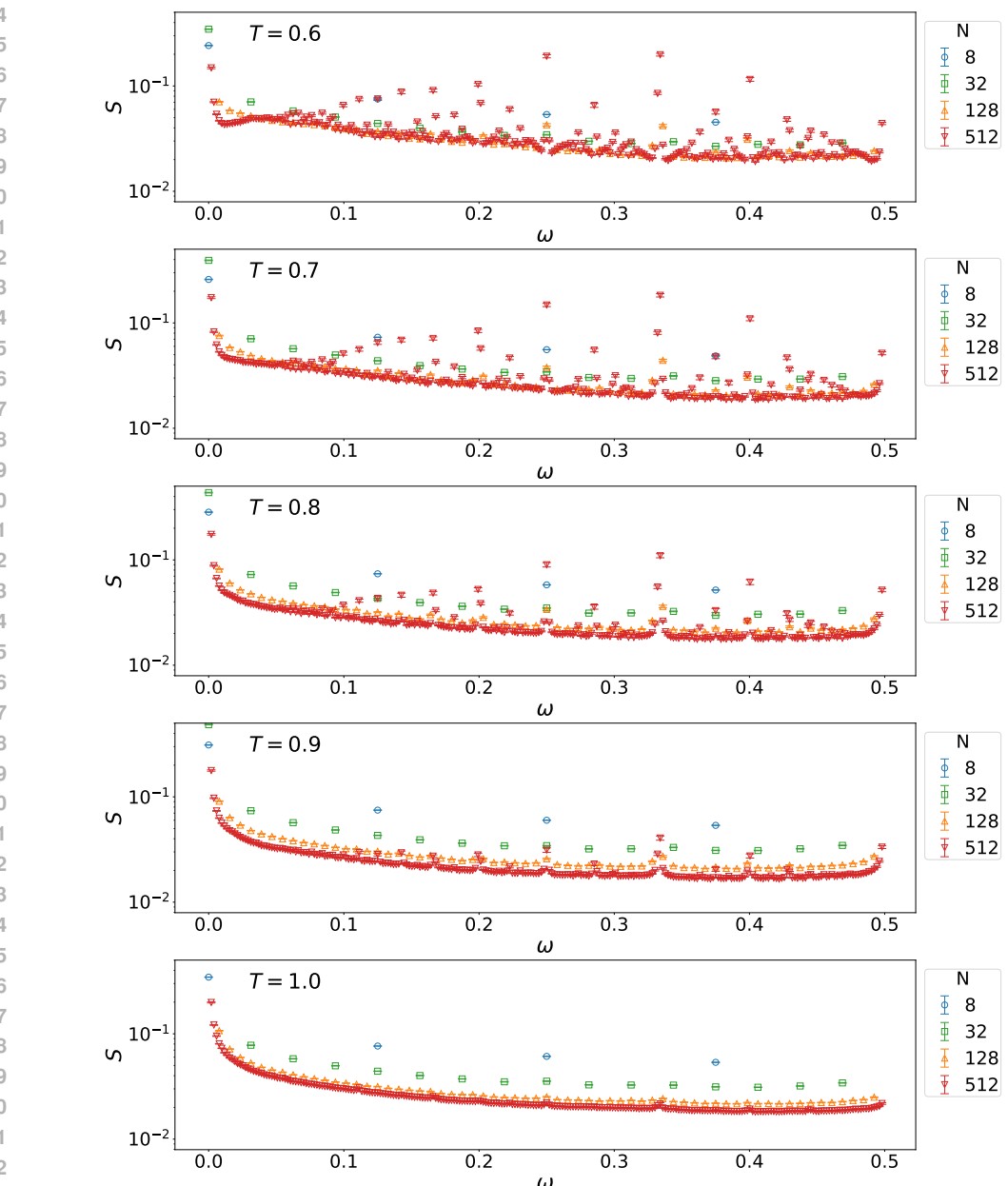

Figure 15: Power spectrum $S = S_{\text{PROPN}}$ as a function of $\omega$ at $T = 0.6, 0.7, 0.8, 0.9$, and $1$.

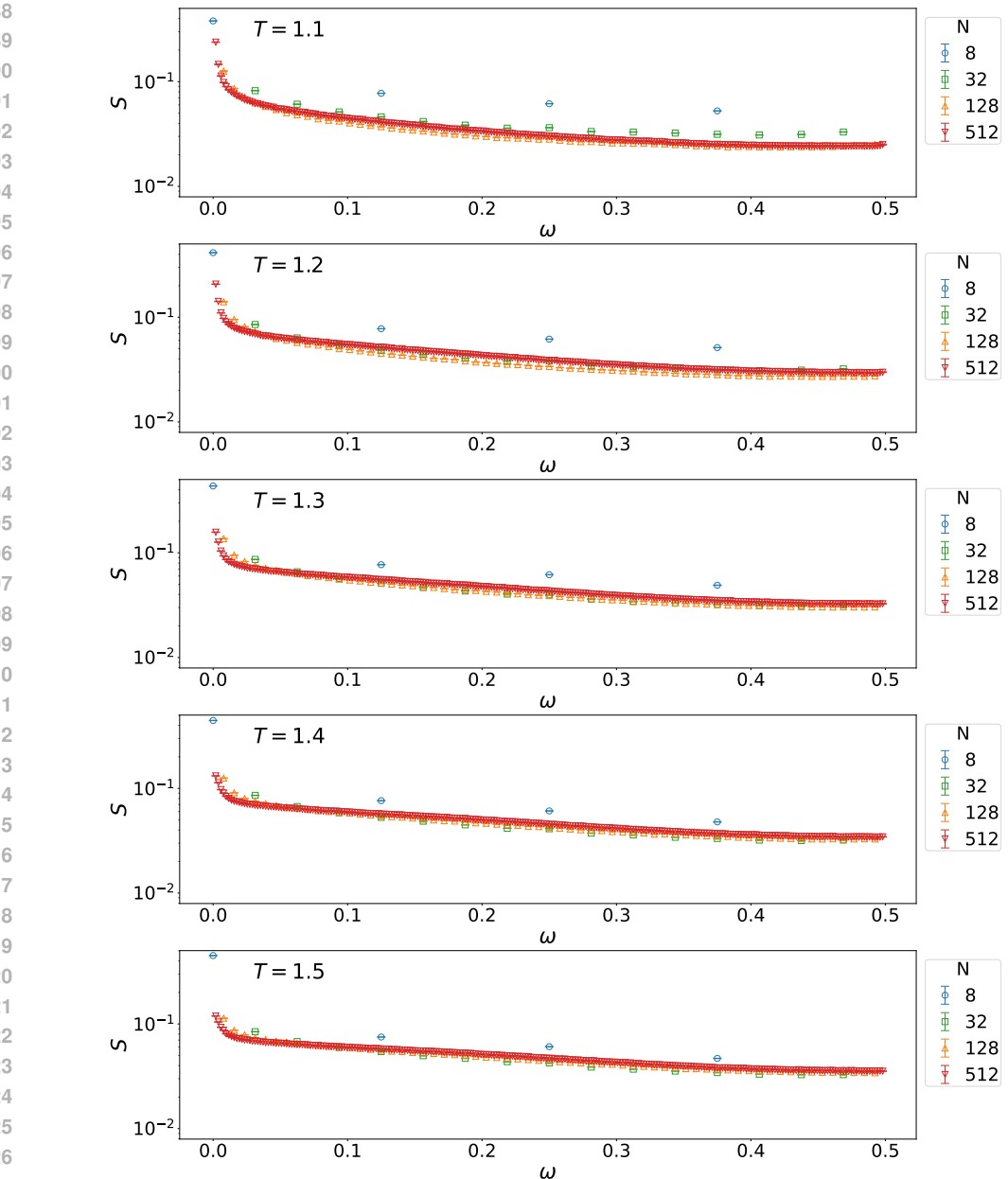

Figure 16: Power spectrum $S = S_{\text{PROPN}}$ as a function of $\omega$ at $T = 1.1, 1.2, 1.3, 1.4,$ and $1.5$.

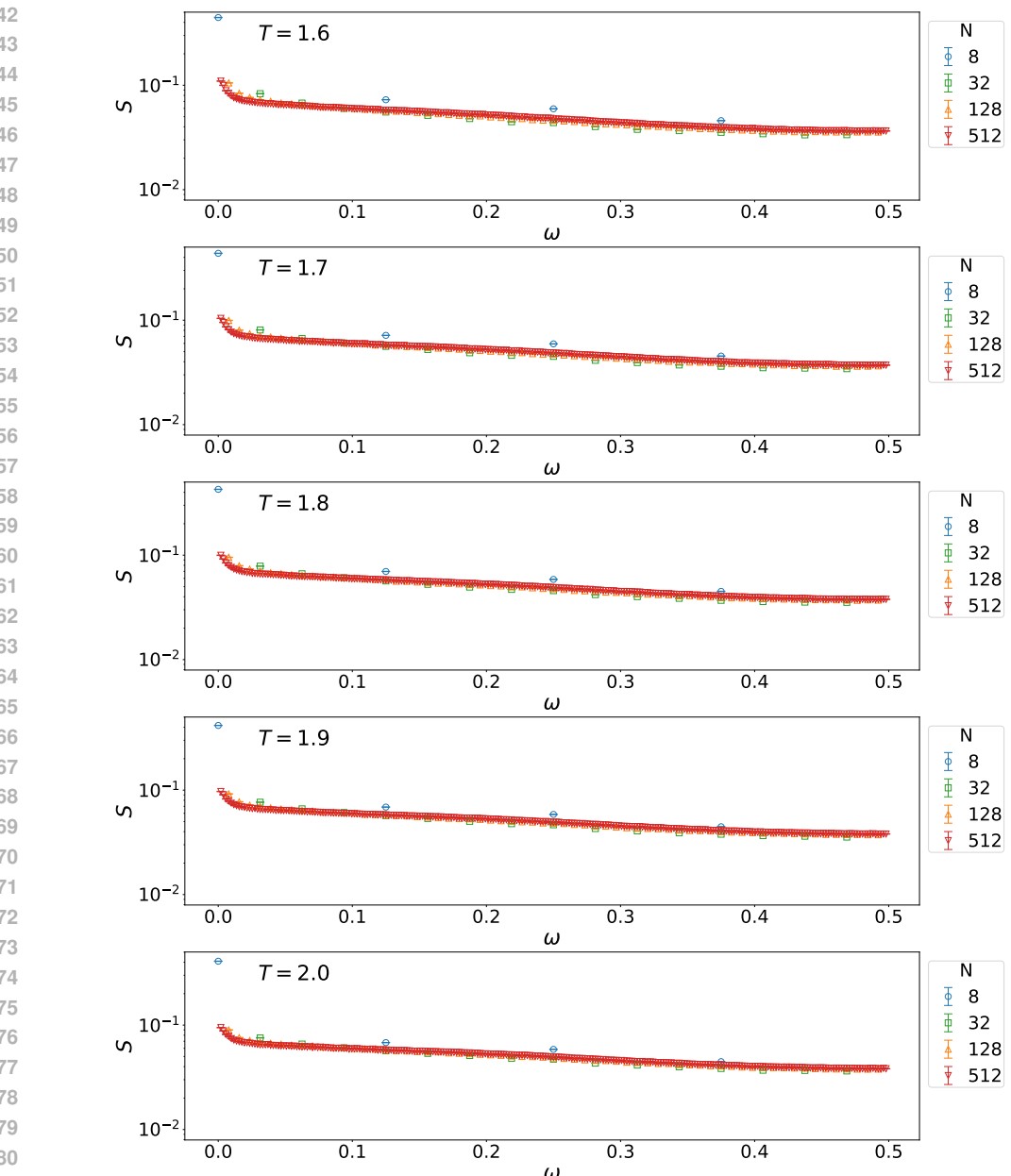

Figure 17: Power spectrum $S = S_{\text{PROPN}}$ as a function of $\omega$ at $T = 1.6, 1.7, 1.8, 1.9$, and 2.

## E    TRANSIENT TIME

As discussed in Sec. 4.3, our results imply that generated texts are natural within the transient time, which diverges as the temperature parameter approaches $T_c$. From a statistical-physical viewpoint, the divergence is expected to follow a power law of the form $\sim |T - T_c|^{-a}$. In general, the exponents characterizing the behavior of statistical quantities near the critical point, such as the exponent $a$, have an essential relationship to the fundamental characteristics of the system. In this appendix, we estimate the transient time and the exponent $a$.

At high temperatures $T > T_c$, although the decay in the correlation presented in Fig. 2 does not exhibit the time scale clearly, we can estimate roughly the scale of the transient by the minimum time interval $\Delta t$ that satisfies $C(0, \Delta t)/C(0,0) < \theta$ with a threshold $\theta$. Figure 18 shows the interval, denoted by $\Delta t_{\text{threshold}}$, as a function of $T - 1$. This result implies that the transient time at high temperatures diverges with an exponent $\approx 1$ when approaching the critical point.

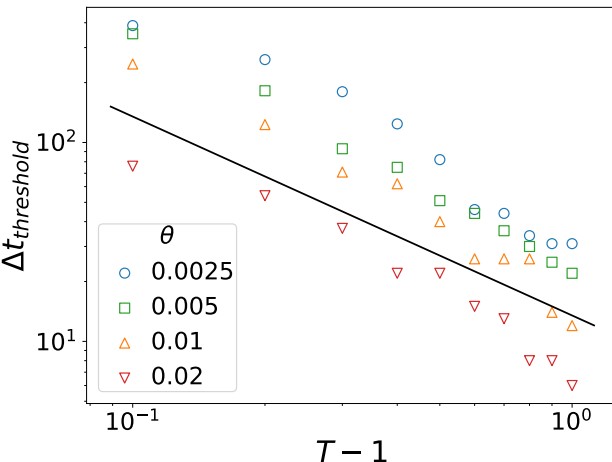

Figure 18: Minimum interval $\Delta t_{\text{threshold}}$ such that the correlation ratio is smaller than the threshold $\theta$ as a function of $T - 1$, for $\theta = 0.0025, 0.005, 0.01$, and $0.02$. The black line represents a line proportional to $(T - 1)^{-1}$.

## F    PRINCIPAL COMPONENT ANALYSIS

In Sec. 4.3, we have discussed the time evolution of the distribution $v_a(t)$ of POS tags, only focusing on $v = v_{\text{PROPN}}$. To intuitively understand the 18-dimensional dynamics, we perform PCA. Specifically, we concatenate $v(0), \cdots, v(N-1)$ at 20 temperature points $T = 0.1, 0.2, \cdots, 2$ into $18 \times 20N$ data matrix and apply PCA to it. The singular values and elements of each principal component are shown in Figs. 19 and 20, respectively. The former suggests that the contributions of PC1 and PC2 are sufficiently large. In the latter, the elements corresponding to PROPN has the largest absolute value in PC1. This justifies that we mainly focus on the dynamics of $v(t)$.

Figure 21 shows the dynamics in $v(t)$ at $T = 0.1, 0.2, \cdots, 2$ projected onto the two-dimensional PC space. These imply that the stationary states that $v(t)$ reaches at $t \to \infty$ can be classified into two types. The dynamics above the critical point converge to one, while that below the point converges to the other. Around the critical point, the transient time is longer than that at higher or lower temperatures. These observations are consistent with the discussion in Sec. 4.3. The dynamics also implies that the slowing down occurs when the two stable states potentially coexist.

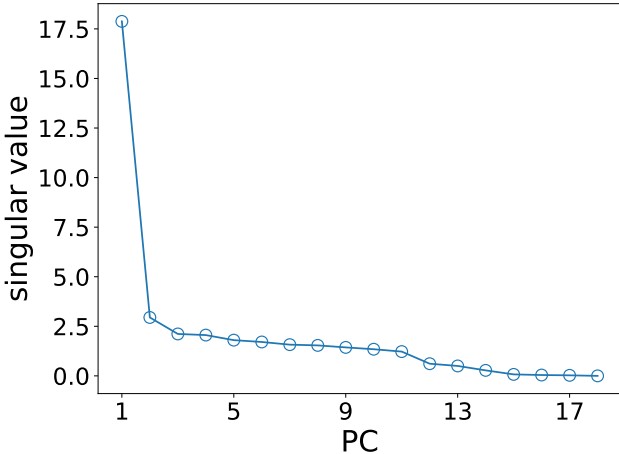

Figure 19: Singular values in PCA on the dynamics in $v(t)$.

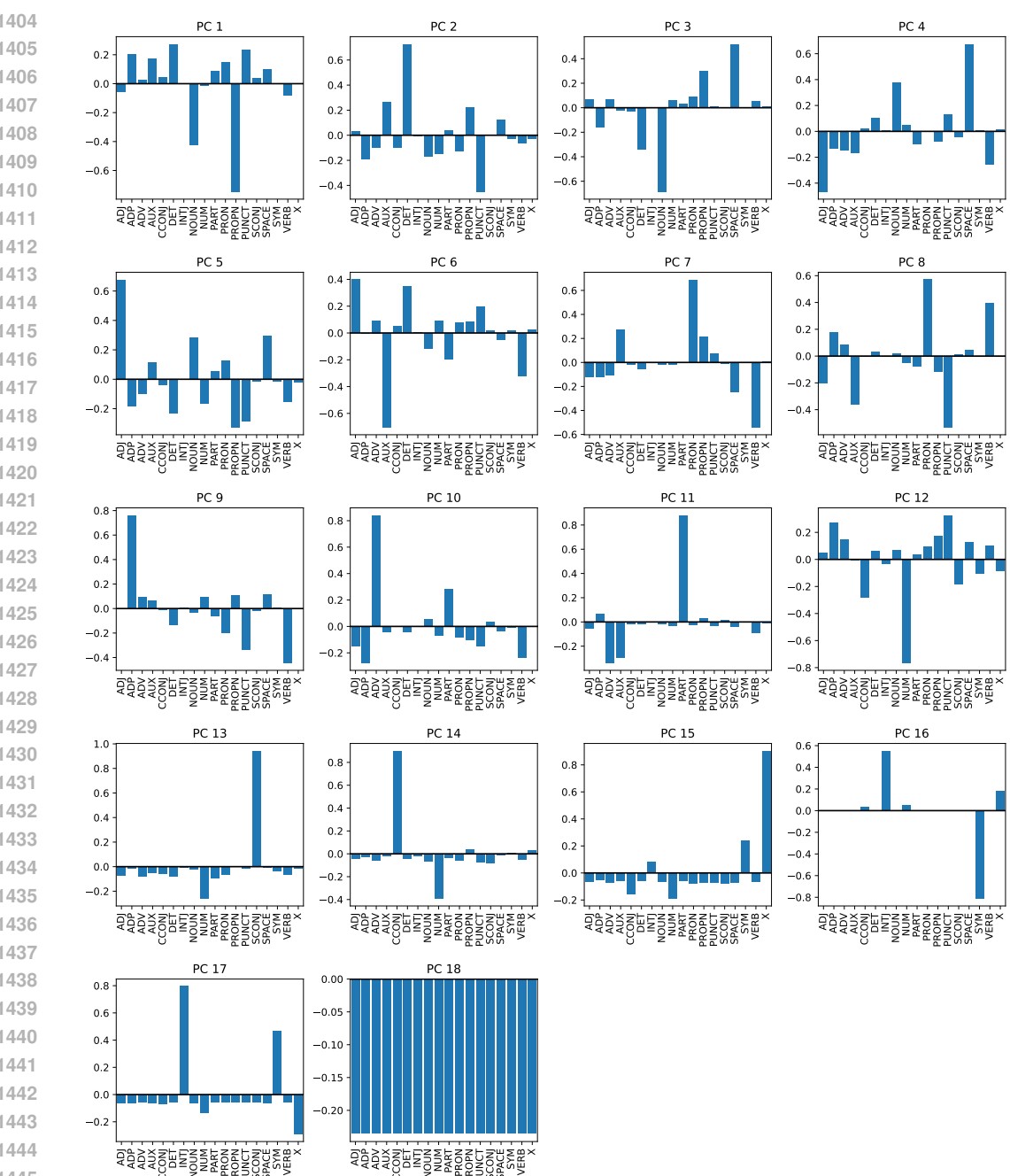

Figure 20: Elements of each principal component in PCA on the dynamics in $\boldsymbol{v}(t)$.

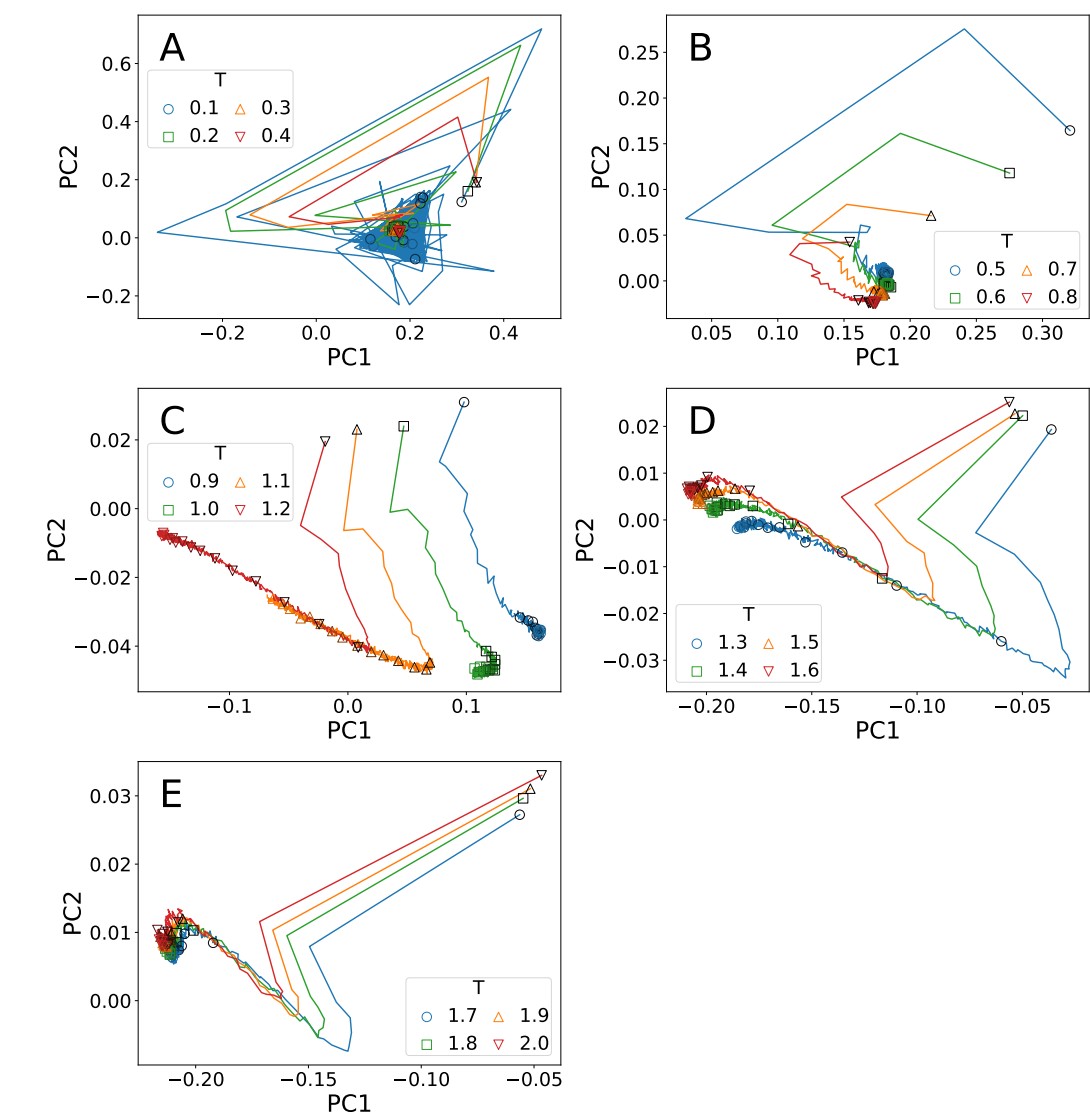

Figure 21: Dynamics in $\boldsymbol{v}(t)$ in sequences generated by GPT-2 at $T = 0.1, 0.2, \cdots, 2$, projected onto the two-dimensional PC space. Markers are plotted every time interval 32. Darker ones represent smaller $t$.

## G  POWER SPECTRA AND TWO-DIMENSIONAL DYNAMICS IN OPENWEBTEXTCORPUS

As we have discussed in Sec. 5, sequences from OWTC share similar properties with critical GPT-2 in correlation, integrated correlation, and the dynamics in $v(t) = v_{\text{PROPN}}(t)$. Here, we show the power spectrum and the dynamics in the two-dimensional PC space in OWTC, which are also similar to those in critical GPT-2.

Figure 22 shows the power spectrum in OWTC for $N = 512$. Those for GPT-2 at $T = 0.9$, 1, and 1.1 are plotted at the same time. Again, the behavior of OWTC is close to that of GPT-2 at $T = 0.9$. However, several peaks exist in the power spectrum in the latter, suggesting the existence of repetitions that is absent in OWTC.

Figure 23 shows that the dynamics in $v(t)$ in OWTC and sequences generated by GPT-2 at $T = 0.9$, 1, and 1.1, projected onto the same two-dimensional PC space as calculated in App. F. We can observe that the dynamics in OWTC follows the trajectory close to that of GPT-2 at $T = 0.9$ for large $t$, similarly to the dynamics in $v(t)$.

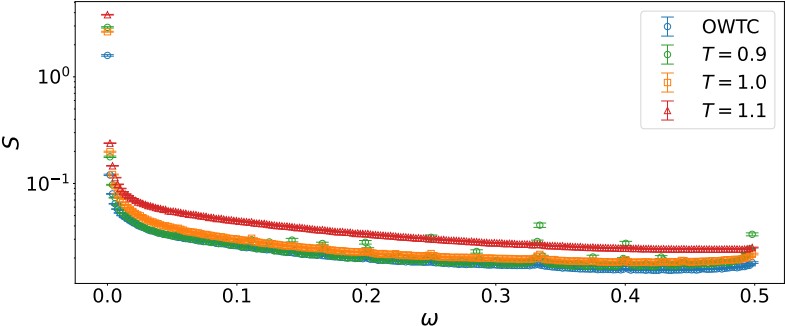

Figure 22: Power spectrum $S = S_{\text{PROPN}}$ as a function of $\omega$ in OWTC and sequences generated by GPT-2 at $T = 0.9$, 1, and 1.1. The sequence length is $N = 512$.

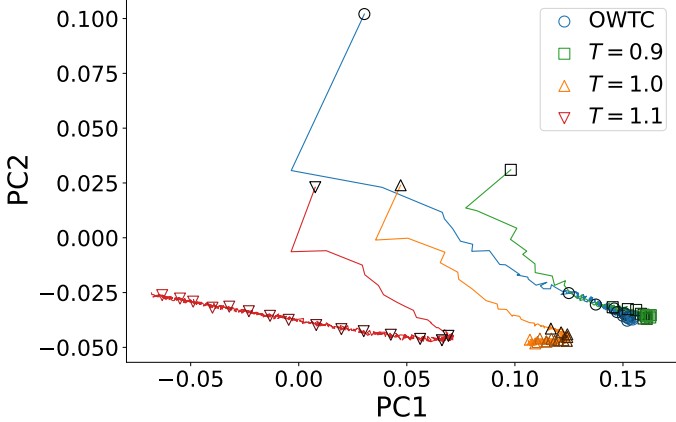

Figure 23: Dynamics in $v(t)$ in OWTC and sequences generated by GPT-2 at $T = 0.9$, 1, and 1.1, projected onto the two-dimensional PC space. Markers are plotted every time interval 32. Darker ones represent smaller $t$.

## H  CRITICAL PROPERTIES OF WIKITEXT

In Sec. 5 and App. G, we have examined the statistical properties of OWTC, such as the perplexity of GPT-2, correlations, power spectra, and dynamics, showing that the corpus is statistically close to critical GPT-2. We perform the same set of analyses on another corpus, wikitext-103-raw-v1 (Merity et al., 2016). The perplexity in Fig. 24 (A) shows that GPT-2 fits the corpus best at $T \approx 1$, similarly to the result for OWTC. In the results of statistical quantities presented in Figs. 24 (B–F), we can see all the critical properties observed in OWTC, such as power-law decay of correlation, divergent integrated correlation, and slow dynamics. This implies that these critical properties can be observed across different natural language datasets.

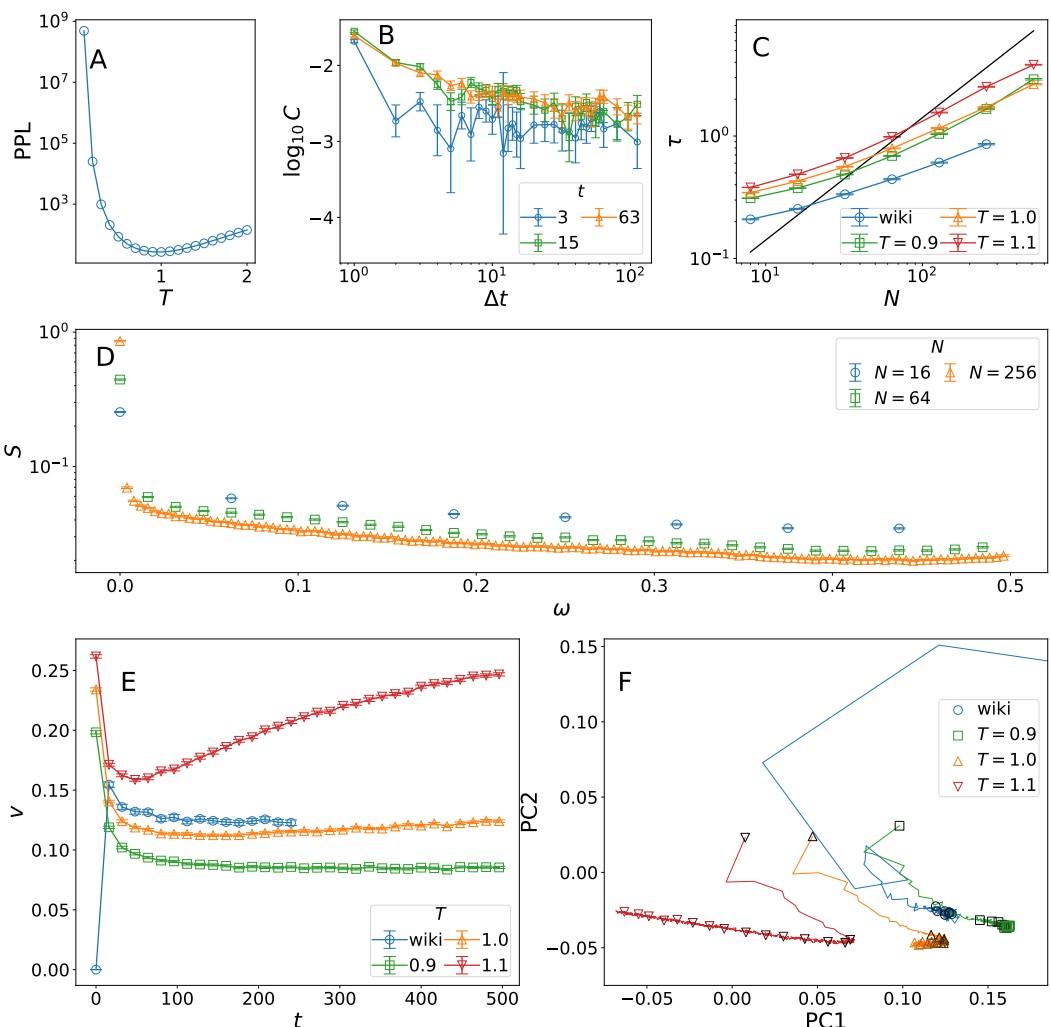

Figure 24: (A) Perplexity of GPT-2 on $10^4$ rows extracted from WikiText with varying temperature $T$. (B) Correlation $C(t, t+\Delta t) = C_{\text{PROPN,PROPN}}(t, t+\Delta t)$ in WikiText as a function of time interval $\Delta t$, where the sequence length is $N = 256$. (C) Integrated correlation $\tau = \tau_{\text{PROPN,PROPN}}$ in WikiText and sequences generated by GPT-2 at $T = 0.9, 1$, and $1.1$, as a function of sequence length $N$. The black line represents a line proportional to $N$. (D) Power spectrum $S = S_{\text{PROPN}}$ as a function of $\omega$ in WikiText. (E) Probability $v(t) = v_{\text{PROPN}}(t)$ that the $t$-th tag is PROPN in WikiText with $N = 256$ and sequences generated by GPT-2 at $T = 0.9, 1$, and $1.1$, with $N = 512$, as a function of time $t$. (F) Dynamics in $\boldsymbol{v}(t)$ in WikiText with $N = 256$ and sequences generated by GPT-2 at $T = 0.9$, $1$, and $1.1$, with $N = 512$, projected onto the two-dimensional PC space. Markers are plotted every time interval 32. Darker ones represent smaller $t$. The results in (B–F) were calculated based on $4 \times 10^4$ POS sequences from WikiText.

# I ANALYSIS OF JAPANESE GPT-2

In the main text, we have investigated statistical properties of POS sequences generated by GPT-2 with varying temperature and have found the phase transition between repetitive sequences and incomprehensible ones. An interesting question is if this phase transition occurs with other languages and LLMs. Here, we analyze Japanese GPT-2 medium with 361M parameters (Zhao & Sawada; Sawada et al., 2024) using ja_core_news_sm pipeline from spaCy library (Montani et al., 2023). For each $T$ and $N$, $2.4 \times 10^3$ sequences were sampled. Error bars in the figures represent an $80\%$ confidence interval estimated using the symmetric bootstrap-$t$ method (Hall, 1988).

As shown in Fig. 25, the integrated correlation diverges at low temperatures, while it seems to converge to a finite value at high temperatures. This result implies the phase transition point, such that the decay of correlation is slow below it and rapid above it. The power spectrum is also shown in Figs. 26–29. From these, we can observe that repetitive structures with many distinct Fourier modes emerge only at low temperatures. These results suggest that this LLM exhibits a phase transition similar to that in GPT-2.

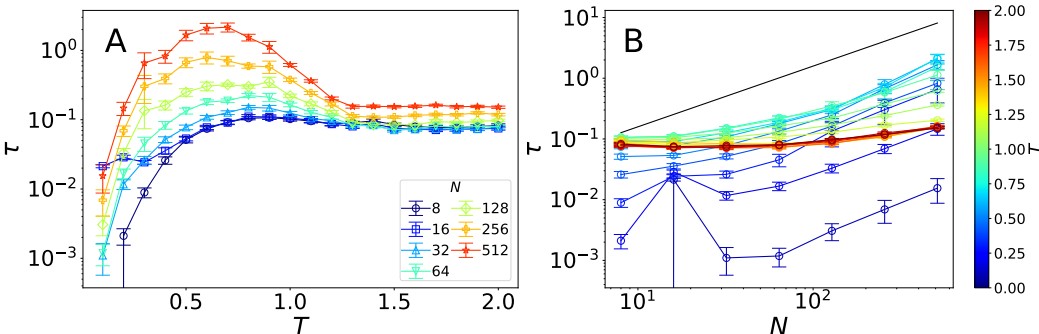

Figure 25: (A) Integrated correlation $\tau = \tau_{\text{PROPN,PROPN}}$ in sequences generated by Japanese GPT-2 as a function of temperature $T$ for various sequence length $N$. (B) The same quantity as a function of sequence length $N$ for various temperatures $T$. The black line represents a line proportional to $N$.

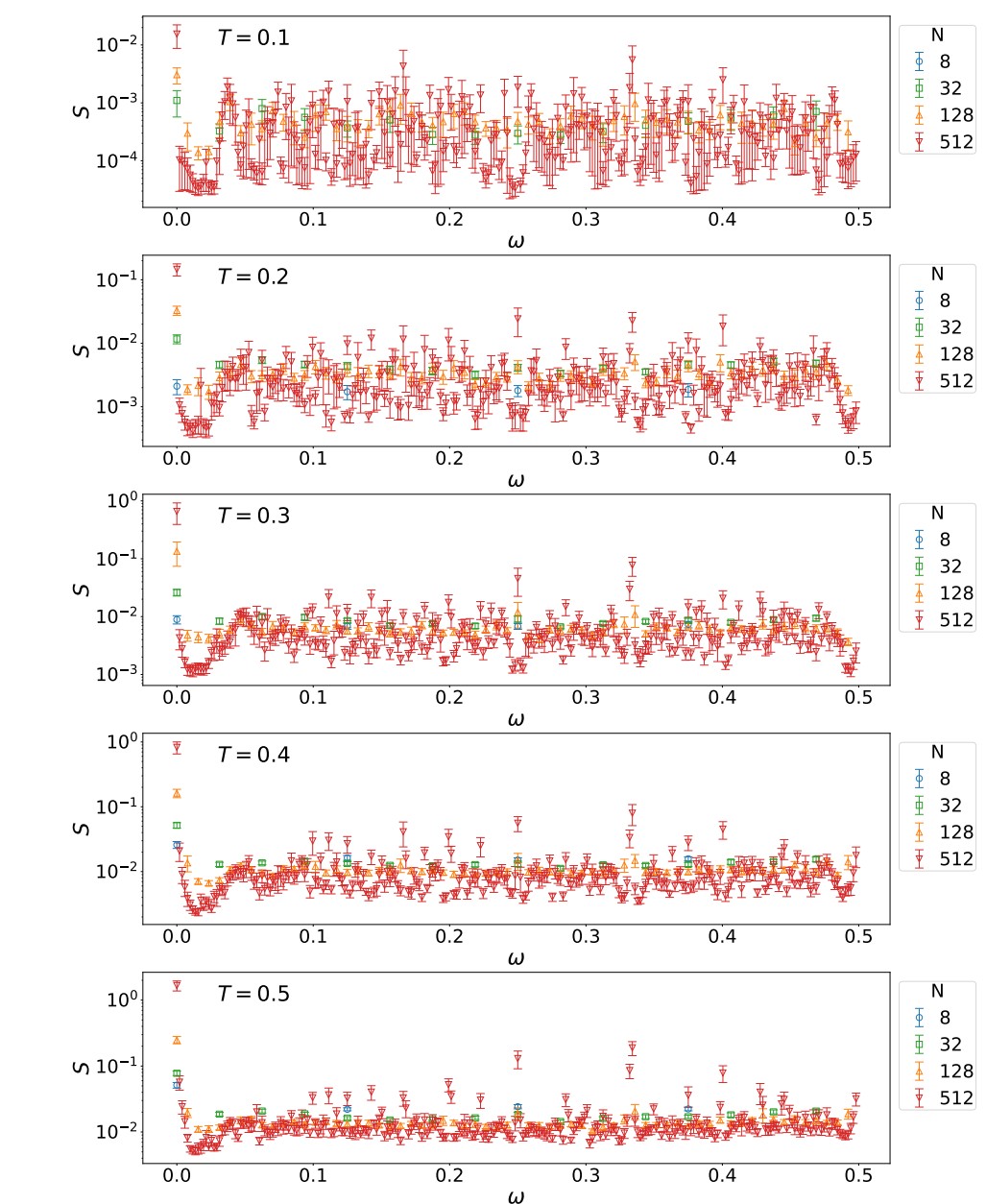

Figure 26: Power spectrum $S = S_{\text{PROPN}}$ in sequences generated by japanese-gpt2-medium as a function of $\omega$ at $T = 0.1, 0.2, 0.3, 0.4,$ and $0.5$.

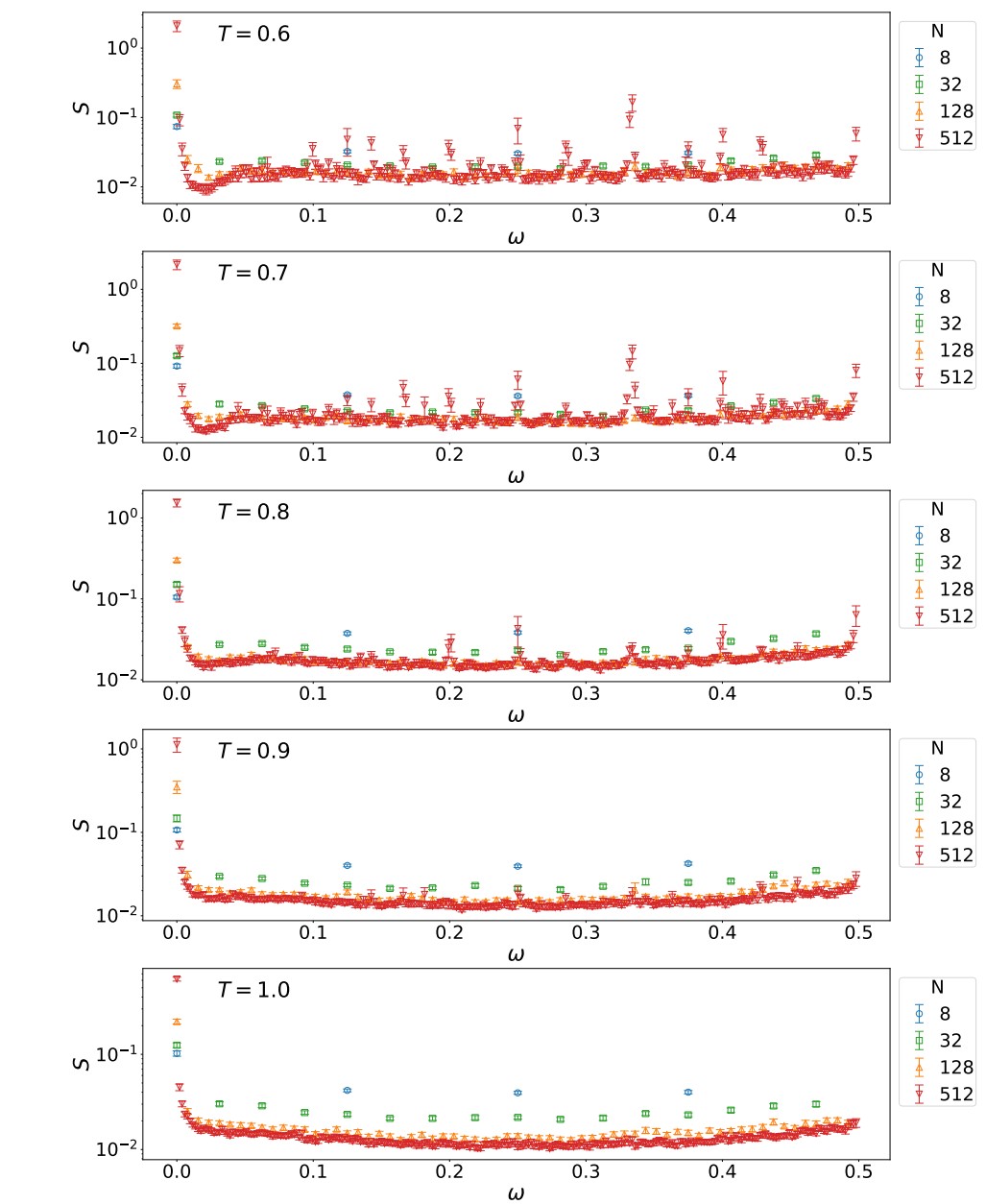

Figure 27: Power spectrum $S = S_{\mathrm{PROPN}}$ in sequences generated by japanese-gpt2-medium as a function of $\omega$ at $T = 0.6, 0.7, 0.8, 0.9,$ and 1.

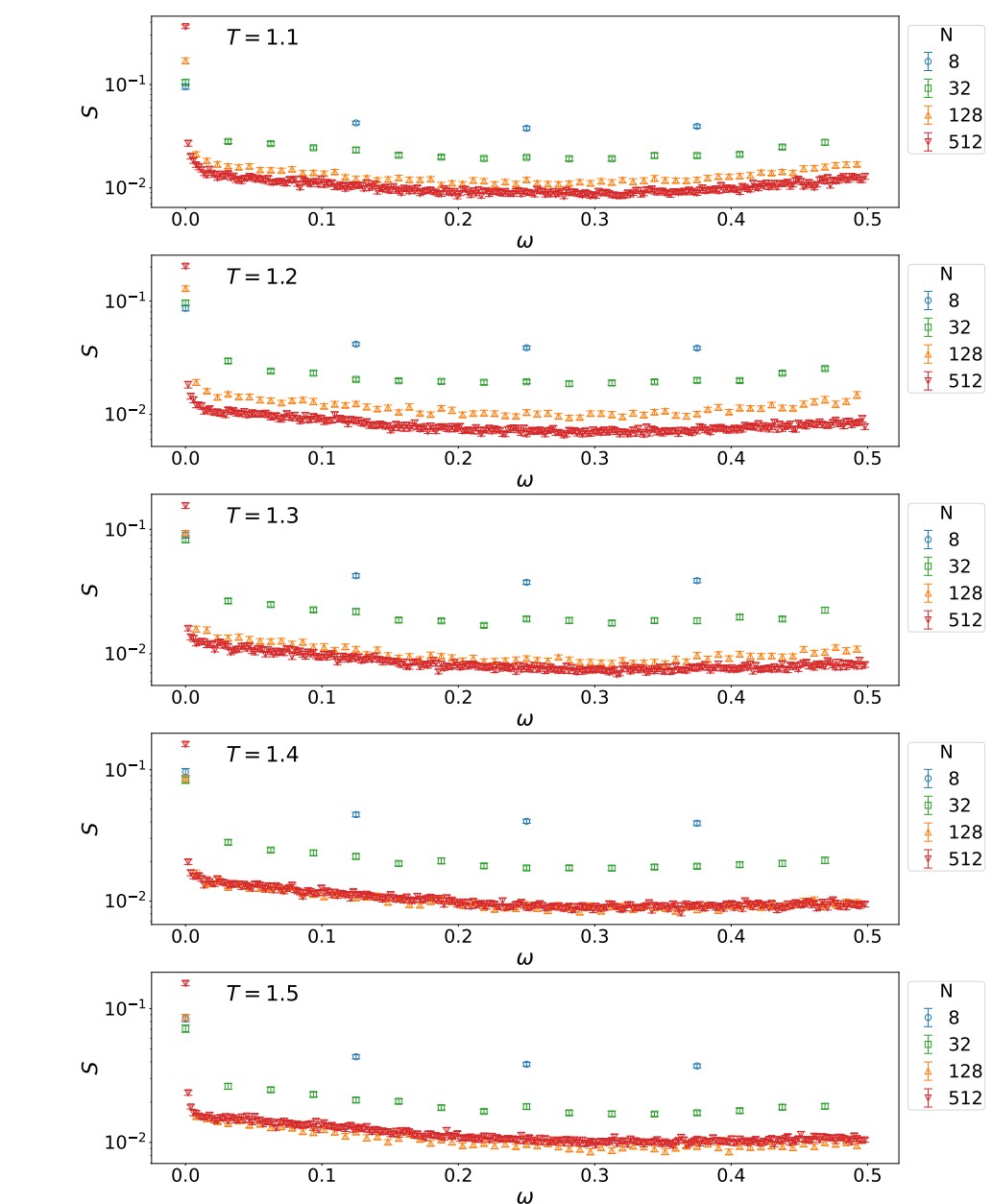

Figure 28: Power spectrum $S = S_{\mathrm{PROPN}}$ in sequences generated by japanese-gpt2-medium as a function of $\omega$ at $T = 1.1, 1.2, 1.3, 1.4$, and $1.5$.

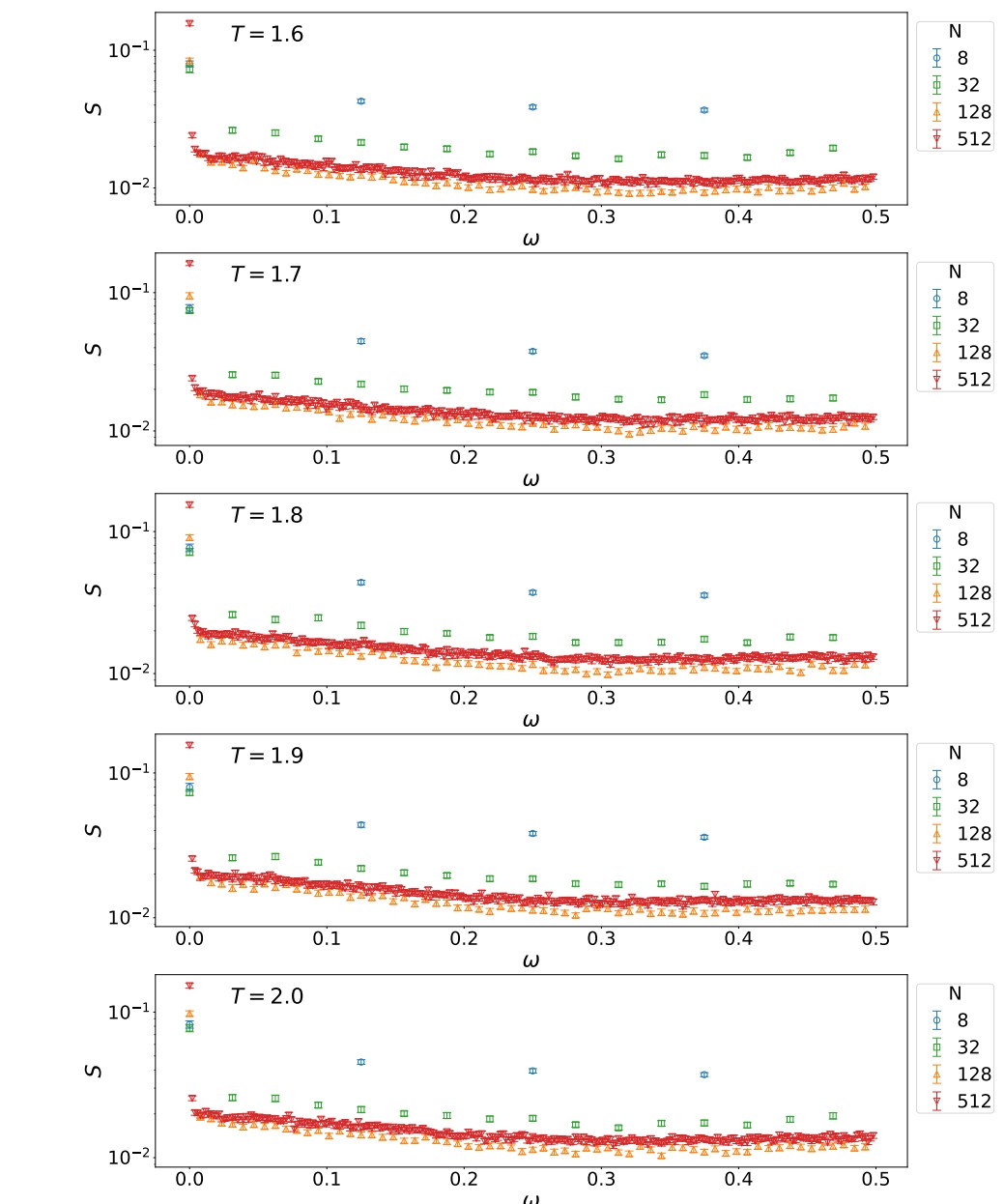

Figure 29: Power spectrum $S = S_{\text{PROPN}}$ in sequences generated by japanese-gpt2-medium as a function of $\omega$ at $T = 1.6, 1.7, 1.8, 1.9$, and $2.0$.

# J ANALYSIS OF PYTHIA

In the main text, we have generated texts with GPT-2 and mapped them to POS sequences, to show the existence of a phase transition. In this appendix, to examine if the phase transition is observed with other models and mappings, we use Pythia with 70M and 1B parameters (Biderman et al., 2023) and a character-based mapping. First, we transformed texts to POS sequences in the same manner as in the main text. Second, we also transformed texts to sequences of numerals, where "A" or "a" is mapped to 0, "B" or "b" is mapped to 1, ... , "Z" or "z" is mapped to 25, other characters to 26, and space, tab, etc. to 27. For each $T$ and $N$, $10^4$ sequences were sampled. Error bars in the figures represent an $80\%$ confidence interval estimated using the symmetric bootstrap-$t$ method (Hall, 1988).

Figures 30 and 31 show the integrated correlations $\tau_{\text{PROPN,PROPN}}$ for POS sequences and $\tau_{26,26}$ for character-based sequences generated by Pythia 70M, respectively. Note that $\tau_{\text{PROPN,PROPN}}$ and $\tau_{26,26}$ have the largest contributions for each setup. Similarly, Figs. 32 and 33 show the corresponding results for sequences generated by Pythia 1B. All these results suggest that the correlation diverges at low temperatures and converges at high temperatures. These results, similar to those in the main paper, support that the phase transition can occur with different mappings and LLMs, including those of larger sizes.

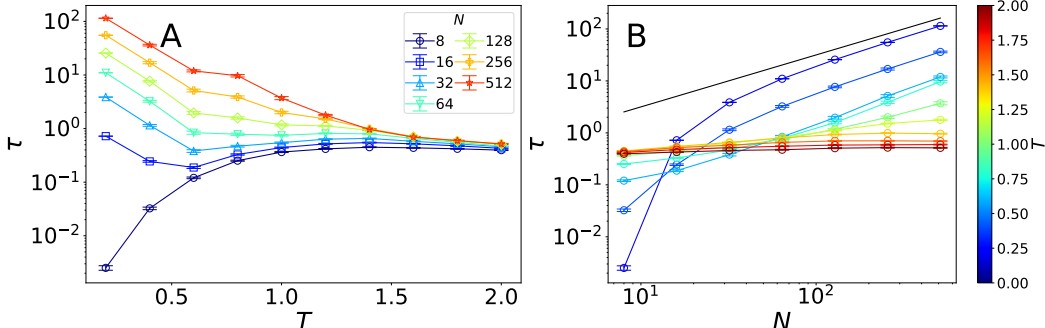

Figure 30: (A) Integrated correlation $\tau = \tau_{\text{PROPN,PROPN}}$ in POS sequences generated by Pythia 70M as a function of temperature $T$. (B) The same quantity as a function of sequence length $N$. The black line represents a line proportional to $N$.

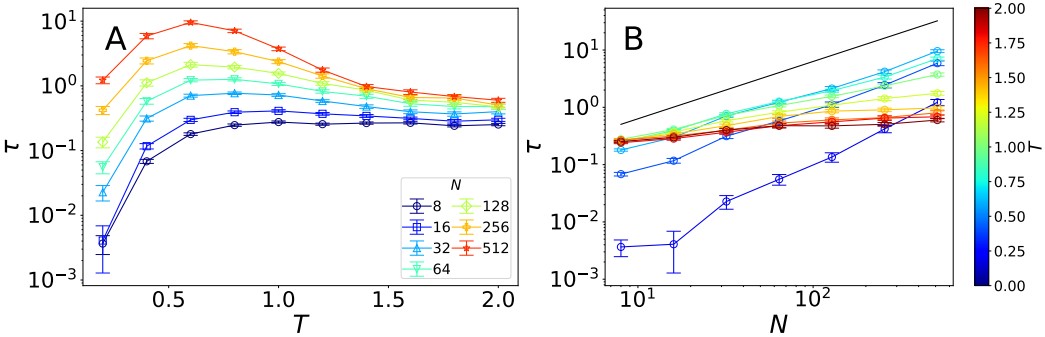

Figure 31: (A) Integrated correlation between 26 (corresponding to other characters) and 26 in numeral sequences obtained by the character-based mapping from texts generated by Pythia 70M, as a function of temperature $T$ (B) The same quantity as a function of sequence length $N$. The black line represents a line proportional to $N$.

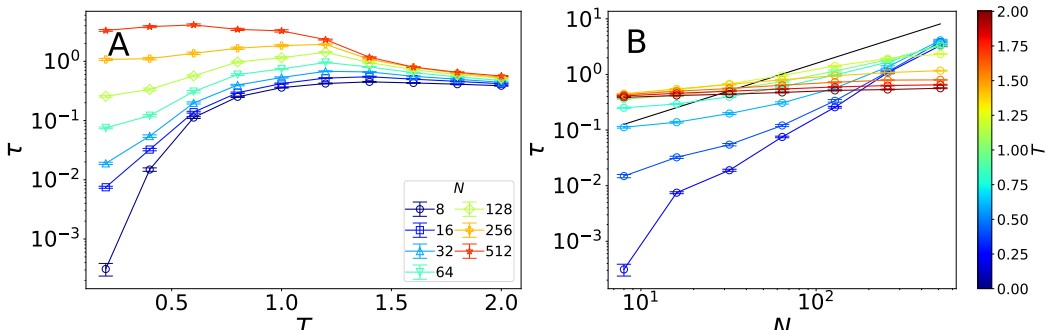

Figure 32: (A) Integrated correlation $\tau = \tau_{\text{PROPN,PROPN}}$ in POS sequences generated by Pythia 1B as a function of temperature $T$. (B) The same quantity as a function of sequence length $N$. The black line represents a line proportional to $N$.

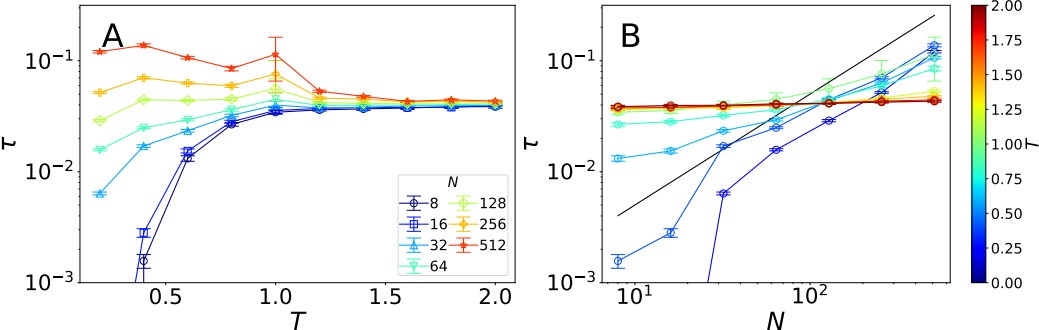

Figure 33: (A) Integrated correlation between 26 (corresponding to other characters) and 26 in numeral sequences obtained by the character-based mapping from texts generated by Pythia 1B, as a function of temperature $T$ (B) The same quantity as a function of sequence length $N$. The black line represents a line proportional to $N$.

