# OpenReview forum: "Critical Phase Transition in Large Language Models"
_ICLR.cc/2025/Conference — Submitted to ICLR 2025_

### Official Review · Reviewer_NkYN · 2024-10-31

**Soundness:** 3
**Presentation:** 3
**Contribution:** 2
**Rating:** 3
**Confidence:** 4

**Summary:**

This paper identifies a phase transition around temperature 1.0 when increasing the temperature parameter for LLM text generation. They focus on POS tags of generated sequences, finding a phase change in (1) the correlation of POS tags at two time points (for some time change $\Delta t$) and (2) periodic behavior. Natural text seems to correspond to roughly temperature 1.0.

**Strengths:**

1. Drawing a connection between statistical physics and LLM text generation is very interesting.
2. The paper identifies a phase transition for multiple metrics (correlation between two tags, power spectra).

**Weaknesses:**

1. The results for periodic behavior are nicely quantified, but slightly unsurprising (degeneration into repeated sequences for low temperature, but not high temperature; e.g. https://arxiv.org/pdf/1904.09751). Looking at the specific types of repetitive loops that the model falls into could be more informative.
2. The result that GPT-2 is closest to natural language text at T=1.0 is a bit unsurprising given that this leaves the probabilities unchanged from how they were trained. For example, we would likely expect lowest perplexity over a large corpus for T=1.0, because the model is trained to minimize loss (log-perplexity) for T=1.0.

**Questions:**

1. What is the motivation behind the POS tag correlation on p. 5? It's not necessarily intuitive why a proper noun appearing at time t should be highly correlated with a proper noun appearing at time $t+\Delta t$ for large $\Delta t$ (e.g. 100 tokens later; Figure 2). What do these correlations tell us about the actual text that gets generated?
2. Consider including the word temperature in the title; phase changes often refer to transitions over the course of pretraining (https://arxiv.org/abs/2209.11895) rather than varying the temperature parameter, so it would be helpful to clarify that this paper focuses on varying temperature.

---

> ### Author Response · Authors · 2024-11-19
>
> We appreciate the reviewer’s thoughtful feedback, which has helped strengthen our discussion. Below, we address each weakness and question.
>
> # Weakness 1
> > The results for periodic behavior are nicely quantified, but slightly unsurprising
>
> We agree that the emergence of repetitive structures at low temperatures is not new. Nevertheless, our major finding is that the change from the low-temperature, repetitive regime to the high-temperature, incomprehensible regime is not a crossover but a phase transition with a singularity. In other words, the boundary between the two regimes is uniquely defined with the divergent statistical quantities. This is unexpected and nontrivial.
>
> > Looking at the specific types of repetitive loops that the model falls into could be more informative.
>
> We also agree that examining specific repetitive structures could offer valuable insights. However, it is also intriguing to observe the power spectrum averaged over all sequences because it reveals that LLMs potentially exhibit countless periodic structures associated with many distinct peaks. This behavior contrasts with typical natural phenomena, where the power spectrum in the ordered phase has only a finite number of peaks. We have clarified this point in the revised manuscript.
>
> # Weakness 2
> > The result that GPT-2 is closest to natural language text at T=1.0 is a bit unsurprising
>
> As noted in Sec. 6, it is reasonable that LLMs align most closely with natural languages at $T\approx 1$. Yet, it is highly nontrivial that a phase transition exists at this point, which is the main focus of our study. Our point of showing the result is that the critical behaviors at the phase transition point should be associated with natural language.
>
> # Question 1
> > What is the motivation behind the POS tag correlation on p. 5?
>
> As we have mentioned in Sec. 4.1, we mainly focus on the correlation between PROPN and PROPN because it has the largest contribution among $18^2$ different correlations. We do not claim the PROPN-PROPN correlation is special. Indeed, similar critical behaviors are observed for other POS pairs, as discussed in App. B.
>
> > What do these correlations tell us about the actual text that gets generated?
>
> Suppose that the PROPN-PROPN correlation decays exponentially with $\Delta t$ as $C \sim \exp(-\Delta t / \xi)$. Then, $C$ is negilible for $\Delta t\gg \xi$. Therefore, once a PROPN appears, another PROPN is likely to appear within the $\xi$ interval, and after that POS tags appear only randomly. By contrast, the actual correlation at $T=1$ decays in a power-law, which is qualitatively slower than exponential decays. In other words, $\xi$ is formally infinite. This implies that the occurrence of PROPN at one point in time can influence events at arbitrarily distant times. Importantly, similar phase transitions are observed between other POS pairs (see App. B) and between characters (see App.J). Additionally, the critical decay in natural languages has been observed across different linguistic units, including words, characters, and phones. This suggests that in both natural languages and LLMs at the critical point, the influence of _any_ event can propagate throughout the entire text. We have emphasized this implication in the revised manuscript.
>
> # Question 2
> > Consider including the word temperature in the title
>
> We thank the reviewer for the suggestion. We decided to change the title to _Temperature-Induced Critical Phase Transition in Large Language Models_.

---

> > ### Comment · Reviewer_NkYN · 2024-11-23
> >
> > Thank you for the response! This answers my questions, but I've decided to keep my score unchanged. I still feel that some of the results are a bit unsurprising, and I agree with reviewer LsMv that the investigation of temperature alone is fairly narrow (an external sampling hyperparameter, as the original model output probabilities remain unchanged).

---

> > > ### Author Response · Authors · 2024-11-28
> > >
> > > We appreciate the reviewer’s additional comments and regret that the significance of our work may not have been fully conveyed. While further research is necessary to establish a comprehensive theory of phase transitions in LLMs, we believe that our research is the basis for future studies that will offer useful insights into their behavior.
> > >
> > > To support this, we have conducted additional analyses as presenteed in App. E. We have estimated the transient time, within which generated texts are expected to be natural. Our results show that the transient time at $T>T_c$ diverges following a power-law function of $T-T_c$ with an exponent  of approximately 1 as $T$ approaches $T_c$.
> > >
> > > With varying other parameters including internal ones, we can estimate similar exponents, which would also characterize the length scale at which texts remain natural. From a statistical-physical viewpoint, these exponents are expected to be related through a universal relation. Furthermore, the set of the exponents is determined by the fundamental characteristics of the system. Therefore, precise estimation of these exponents with varying different parameters would offer theoretical, quantitative insights into the parameter regions where LLMs generate natural texts.

---

### Official Review · Reviewer_HR6F · 2024-11-04

**Soundness:** 3
**Presentation:** 4
**Contribution:** 3
**Rating:** 6
**Confidence:** 3

**Summary:**

This paper investigates whether phase transitions occur in Large Language Models (LLMs). The authors conduct a statistical analysis of sequences generated by GPT-2 at various temperatures and found the phase transition occurs.

**Strengths:**

1. The project fosters the understanding of LLMs from an interesting angle of phase transition.
2. The experiments are well designed and cleared documented.

**Weaknesses:**

1. While LLMs develop fast over the past few years, this work tests on GPT-2, instead of newer generations of models.

**Questions:**

1. In what ways will the understanding of phase transition impact future LLM works?

---

> ### Author Response · Authors · 2024-11-19
>
> We appreciate the reviewer’s positive evaluation of our manuscript. Below, we address the weakness and question raised by the reviewer.
>
> # Weakness
> > While LLMs develop fast over the past few years, this work tests on GPT-2, instead of newer generations of models.
>
> We have added the results for Pythia 1B in the appendix. Similar phase transitions are observed in models of 70M, 124M, 361M, and 1B, suggesting that these phenomena also occur in even larger models.
>
> # Question
> > In what ways will the understanding of phase transition impact future LLM works?
>
> The correlation in texts decays according to a power law across different corpora and languages, suggesting that the critical properties are a necessary condition for texts to be natural. The understanding of the mehcanism of the phase transition and critical phenomena will provide insights into how LLMs emulate the critical poroperties of natural languages.
>
> Furthermore, existing statistical-mechanical models cannot exhibit phase transitions with the unique properties observed in this study, such as the complex repetitive structures in the low-temperature regime and the non-exponential decay of correlations in the high-temperature regime. This suggests that existing statistical-mechanical analyses fail to fully capture the nature of LLMs. An important direction for future research is to explore the development of mathematical models that exhibit phase transitions similar to those observed in this study.
>
> Another interesting direction is to identify the universality classes of different models, as discussed in Sec. 6. Systems within the same class share fundamental characteristics, such as dimensionality and symmetry. Therefore, exploring the relationship between the universality classes of different LLMs and their performance would be meaningful.
>
> Finally, the analyses in our study can be applied to changes in LLMs induced by parameters other than the temperature parameter. Therefore, other qualitative changes in LLMs may also be characterized and analyzed from a physical perspective.
>
> We have emphasized these key points in the revised manuscript.

---

> > ### Comment · Reviewer_HR6F · 2024-12-02
> >
> > Thanks for the detailed response! I am keeping the positive score unchanged.

---

### Official Review · Reviewer_7ukK · 2024-11-04

**Soundness:** 2
**Presentation:** 2
**Contribution:** 2
**Rating:** 6
**Confidence:** 2

**Summary:**

This paper argues that LM generations undergo a phase transition at temperature $T = 1$. Specifically, they generate texts from GPT-2 using different temperatures (conditioned on only the BOS token), and map each text into a sequence of POS tags (though the focus only on PROPN), and study three features of the POS tags.

1. **Correlation between two tags:** The authors study whether PROPN tags are correlated at particular intervals in the text. They find the integrated correlation $\\tau$ between PROPN and PROPN saturates to a finite value for $T>1.0$, which means correlation decays faster than critical decay, whereas for $T<1.0$, $\\tau$ diverges, which means the correlation converges to a finite value. In other words, sequences with $T<1$ have long-range correlation between POS tags, and sequences above $T>1$ converge to zero correlation.
2. **Power spectra of POS sequences**: The authors study whether there is periodic appearance of the PROPN tag via the power spectrum. Again, there appears to be long-range order at $T<1.0$, but at $T>1.0$ the spectrum is featureless.
3. **Time evolution of POS sequences:** The authors study the probability of the PROPN tag at time $t$. They find that this probability converges quickly to a limiting value for $T<1.0$ and $T>1.0$, but converges much more slowly for values close to $T\approx 1.0$.

**Strengths:**

This paper investigates whether LLMs go through phase transition with the temperature parameter. This (I believe) contrasts with much of the literature, which is about the possibility of a phase transition with the size of the model. The statistics that the authors introduce may have value for future work that aims to study structural features of LLM generations.

The writing and organization are clear, and the figures are clearly explained.

**Weaknesses:**

- There are obvious empirical concerns: namely, in the main paper, the authors only study GPT-2 small, and all of the analysis is specifically about the structure of where the proper noun PROPN tag occurs in generated text. This must be quite narrow, and it would be more convincing if the authors could summarize the results for other models and other POS tags in the main paper (they are referenced as being in the Appendix, which I did not read).
- This is very far from my area of expertise, but I am not convinced that there is truly *singularity* at $T=1.0$. In their analysis in §4.2 (power spectra) and §4.3 (time evolution), the model seems to demonstrate continuous behavior around $T=1.0$. In particular, I am looking at Figure 3 (A), where the peak in integrated correlation actually occurs at $T=1.1$, and Figure 5, where the values neighboring $T=1.0$ (from $0.9$ to $1.2$) all behave qualitatively similarly to $T=1.0$. This makes me worried that the paper is just showing that generations have structurally different qualities for small $T$ and large $T$, which is somewhat obviously true because $T$ directly controls the peakiness of the LM prediction, such that low $T$ will be highly redundant and high $T$ will be quite degenerate (esp. for GPT-2 era models).
- It is not clear to me how important the authors' findings are for the community. They mention their results suggest "a meaningful analogy between LLMs and natural phenomena," but the analogy was not clear to me from reading the paper. Are there practical insights about what LMs are learning or how we should build future LMs?
- There seems to be [related work](https://community.wolfram.com/groups/-/m/t/2958851) from 2023 that also studies phase transition at $T=1.0$, which the authors mention, and the distinction in their contributions is not clear to me. This may lessen the novelty of their work.

**Questions:**

- Can you help me interpret Equation (1)? My understanding is that $a$ and $b$ are POS tags, and $y_t$ is the particular POS tag at position $t$. Therefore, this is measuring whether there is a predictable relationship that when $y_t=a$, then $y_{t+\\Delta t}=b$. What is the purpose of the second term? Does it serve as some kind of normalization, to account for the base frequency of $y_t=a$ and $y_{t+\\Delta t}=b$? Why are there $18 \\times 17/2$ possible pairs $a$ and $b$? I would have assumed $18^2$?
- L. 240: "The plateau value increases with the position $t$ of the former tag." Is this true? It looks like the green line ($t=15$) is ordered between purple ($t=0$) and blue ($t=3$).
- L. 244: What is "the prefactor" of the decay? I am again failing to see a reliable trend for $t$, since it seems like $t=0$, $t=63$, and $t=225$ all have extreme error bars, while $t=3$ and $t=15$ do not.
- L. 301: What does it mean for a spectrum to be featureless?
- For the power spectra analysis, I can see how the plots at different temperatures look different, but why is there *singular* behavior at $T=1.0$?
- In Figure 5, doesn't $v(t)$ immediately reach its limiting value for $T=1.0$, contrary to what's written in the paper?

---

> ### Author Response · Authors · 2024-11-19
>
> We thank the reviewer for useful suggestions. In response to the reviewer’s comments, we have revised the manuscript. In the following, we address each weakness and question.
> # Weakness 1
> > it would be more convincing if the authors could summarize the results for other models and other POS tags in the main paper
>
> We agree with this suggestion. We have added a brief summary of the Appendix in the main paper.
>
> # Weakness 2
> > I am not convinced that there is truly _singularity_ at $T=1.0$
>
> Phase transitions are defined by a singularity (i.e., a divergence) in the large size limit, corresponding to the $N\to\infty$ limit in our case. Fig. 3 shows that the integrated correlation keeps growing with $N$ at $T\lesssim 1$, while it seems to saturate at $T\gtrsim 1$. This strongly suggests that there exists a temperature $T_c\approx 1$, such that the integrated correlation at $N\to\infty$ diverges below $T_c$ and remains finite above it. Thus, a singularity must lie at $T_c$.
>
> Note that we are not claiming that the critical point is exactly at $T_c=1$. Also, for finite $N$, the peak of the integrated correlation is rounded and often deviates from the exact $T_c$ at $N \to \infty$. This _finite-size_ effect generally appears in any system with a phase transition.
>
> # Weakness 3
> > Are there practical insights about what LMs are learning or how we should build future LMs?
>
> Since the power-law decay in correlation is observed across various corpora and languages, critical properties should be a necessary condition for texts to be natural. We expect that LLMs near $T_c$ can emulate these properties during the transient time before reaching the stationary state, while the transient timescale diverges approaching $T_c$. The mechanism of the phase transition should be closely related to how LLMs learn these properties. Although our results do not yet provide concrete implications for the practical development of LLMs, our novel approach should offer a potential theoretical pathway for future advancements. We have emphasized this point in the revised manuscript.
>
> # Weakness 4
> > There seems to be related work from 2023
>
> Bahamondes (2023) claims a phase-transition-like change at $T\approx 0.1$, which is significantly lower than our estimate on the transition temperature $T\approx 1$. Thus, most probably, the work observed something different from what we have studied. Furthermore, the work did not analyze any size dependence in statistical quantities, which is essential for discussing phase transitions defined in the large size limit. Its claim of a phase transition, therefore, remains speculative.
> # Question 1
> > What is the purpose of the second term?
>
> We define the correlation so that it is zero when $y_t$ and $y_{t+\Delta t}$ are statistically independent. More specifically, if $y_t$ and $y_{t+\Delta t}$ are independent of each other, then the correlation is zero because $E[\delta_{a,y_t}\delta_{b,y_{t+\Delta t}}] = E[\delta_{a,y_t}]E[\delta_{b,y_{t+\Delta t}}]$. We have added this explanation to the manuscript.
> > Why are there $18\times 17/2$ possible pairs $a$ and $b$? I would have assumed $18^2$?
>
> The reviewer is right that there are $18^2$ indeed different $C_{ab}$. We have corrected the number in our manuscript.
> # Question 2
> > "The plateau value increases with the position $t$ of the former tag." Is this true?
>
> We intended to convey that the plateau _tends to_ increase with $t$. Because this does not impact our main claim, we have removed this imprecise wording.
> # Question 3
> > What is "the prefactor" of the decay?
>
> When the decay follows $\approx A (\Delta t)^{-\alpha}$, we refer to $A$ as the _prefactor_. Although this tendency is not immediately apparent, the integrated correlation indicates that the decay at high temperatures is indeed faster than the critical decay. In the revised manuscript, we do not use this confusing terminology.
> # Question 4
> > What does it mean for a spectrum to be featureless?
>
> We mean that the power spectrum at high temperatures does not have peaks, suggesting that generated texts lack nontrivial structure. We have revised the manuscript to avoid the use of this vague terminology.
> # Question 5
> > why is there singular behavior at $T=1.0$?
>
> Similarly to the integrated correlation, the power spectrum should be singular at $T_c\approx 1$ as a function of $T$ in the limit $N\to\infty$. Fig. 14 shows that some peaks grow with $N$ for $T\lesssim 1$, whereas no such peaks are observed for $T\gtrsim 1$. Therefore, in the $N\to\infty$ limit, some peaks diverge below $T_c$ and disappear above it. Note again that we do not claim $T_c$ is precisely equal to 1.
> # Question 6
> > In Figure 5, doesn't $v(t)$ immediately reach its limiting value for $T=1.0$
>
> It indeed seems to have converged to a limiting value at $T=1.0$. However, it actually keeps growing very slowly and has not saturated up to $N\leq 512$. This behavior can be seen more clearly in Fig. 6(D).

---

> > ### Comment · Reviewer_7ukK · 2024-11-26
> >
> > I have raised my score as the authors addressed all of my concerns, though my general confidence in reviewing this paper is still low.
> >
> > Although the paper is quite niche, studies only a very old model, and does not have immediate practical value, I believe it is still an interesting research question with a unique experimental design, and would thus increase the diversity of work presented at ICLR.
> >
> > By the way, I agree with another reviewer that "temperature" should definitely be in the revised paper title.

---

> > > ### Author Response · Authors · 2024-11-28
> > >
> > > We sincerely thank the reviewer for raising the score and for acknowledging the uniqueness and potential contributions of our research. If there are any points in our manuscript that remain unclear and contribute to your low confidence, please let us know. We would be glad to provide further clarification.
> > >
> > > Additionally, as in App.E, we have added the results of estimation of the transient time. As we discuss in Sec. 5, this corresponds to the length scale at which generated texts are natural. We have also estimated the exponent characterizing the divergence of transient time. In general, such exponents are crucial for studying critical phenomena, because they are expected to follow a universal relation, and their values are closely related to the fundamental characteristics of the system.

---

### Official Review · Reviewer_LsMv · 2024-11-07

**Soundness:** 2
**Presentation:** 2
**Contribution:** 2
**Rating:** 3
**Confidence:** 4

**Summary:**

This paper proposes that qualitative changes in language model outputs can be analyzed as phase transitions—an analogy drawn from statistical physics. The authors empirically explore the shifts in characteristics of model-generated natural language that appear with sampling temperature scaling, which they posit may resemble phase transitions observed in physical systems. They observe a shift from what they characterize as an ordered to a disordered phases (i.e., the temporal structures in the different sets of sequences are statistically distinctive) as a result of changes in temperature. They conclude that LMs exhibit phase transition-like behavior around critical parameter values. They compare several quantitive attributes of the sequences generated at different temperatures to those of human generated text and find notable differences.

**Strengths:**

* The motivation behind the paper is nice: the application of a well-studied concept from physics can perhaps allow us to use knowledge about/properties of that concept to better understand natural language and language models
* The work provides a comparison of quantitative aspects of human- and machine-generated language, an approach that is more objective than the qualitative comparisons that are often done and thus perhaps a better ground from which to draw conclusions
* The finding that the statistical properties of language at different sampling temperatures is interesting (anecdotally, at least)

**Weaknesses:**

* The paper is generally difficult to follow:
    * There is confusing terminology that isnt defined/contextualized before it is used, e.g., “long-range correlation” in the introduction). This will likely confuse most readers (myself included)
    * The implications of the observed critical properties are incredibly unclear. See subsequent questions for the parts that felt particularly unclear to me, although this is not comprehensive. As such, it is difficult to draw meaningful conclusions from the paper
* The work only looks at changes in model behavior as a result of changing the temperature parameter. This is a very narrow exploration and also somewhat contrived since it's an external change to the model that doesn’t have any implications about what happens when internal model attributes vary
* Only rather small models are used (on the scale of 100M parameters) and so results are not relevant for modern LLMs

**Questions:**

* In the introduction, the motivation that “the decay of correlation in natural languages follows a power law” is used. This is an incredibly vague statement. What are some concrete correlations in natural language that this applies to?
* Terminological clarifications:
    * What are “changes with a singularity”, both in a physics and POS sequence context?
    * What is a concrete example of a ‘critical phenomenon’ in a physical system and what might be an example of one in natural language?
    * What does it mean that “the high-temperature phase is not simply disordered”? What does this look like qualitatively in language? Does it tell us something about human-generated natural language? And what does it mean about a language model?
* What are the implications of causality being broken for the conclusions that can be drawn? It seems as though this might invalidate results. I don’t see why the effect “shouldn’t matter”. I also don’t think it’s accurate to use the example of next characters as a situation where there are not dependency relationships. Anecdotally: language modeling can also be successful at the character-level. Perhaps I am not understanding something.

---

> ### Author Response · Authors · 2024-11-19
>
> We appreciate the reviewer's constructive comments and will address each point below. Please let us know if there are any unclear points, as this will help us further improve our manuscript.
>
> # Weakness 1
> > There is confusing terminology that isnt defined/contextualized before it is used.
>
> We thank the reviewer for highlighting this issue. In the revised version, we have minimized using undefined or ambiguous terminology.
> # Weakness 2
> > The work only looks at changes in model behavior as a result of changing the temperature parameter
>
> We agree that examining additional parameters would provide further insights into LLMs and phase transitions. Although this is outside the scope of the current paper, our analysis is applicable to various parameters, including internal ones. We propose a novel, generic approach to studying changes in LLMs.
> # Weakness 3
> > Only rather small models are used (on the scale of 100M parameters) and so results are not relevant for modern LLMs
>
> We have added results for Pythia 1B in the Appendix. Models of different sizes (70M, 124M, 361M, and 1B) exhibit similar phase transitions, suggesting that this phenomenon occurs for larger models.
>
> # Question 1
> > What are some concrete correlations in natural language that this applies to?
>
> Earlier studies we cite in our manuscript investigate the correlation function or the mutual information between words, characters, phones, and so on. Across these metrics, the power-law decay is consistently observed. We have added this explanation to our manuscript.
>
> # Question 2
> > What are “changes with a singularity”, both in a physics and POS sequence context?
>
> We use the term _singularity_ in the mathematical sense: if a function or its derivative diverges at a point, the function has a singularity at the point. In our study, the integrated correlation diverges below $T_c$ and remains finite above it, meaning a singularity at $T_c$. In the revised manuscript, we have explained the definition of singularity.
>
> > What is a concrete example of a ‘critical phenomenon’ in a physical system and what might be an example of one in natural language?
>
> A typical example in physics showing a critical phenomenon is ferromagnets: Ferromagnets are not magnetized above the critical temperature, and they gradually become magnetized if one applies an external magnetic field. However, at the critical temperature, ferromagnets are highly susceptible to a magnetic field. The susceptibility characterizing the response to the magnetic field is divergent there. In LLMs at the critical point or in natural languages, as we show in the manuscript, the integrated correlation, equivalent to the susceptibility, is expected to diverge at $N\to\infty$. This means, for instance, that a single word or POS tag change can significantly influence the overall structure of a text. We have explained the implication of critical phenomena in ferromagnets and LLMs more clearly in the revised manuscript.
>
> > What does it mean that “the high-temperature phase is not simply disordered”?
>
> In typical physical systems, the correlation in disordered phases decays exponentially. We refer to this behavior as _simply disordered_. In contrast, in the high-temperature phase observed in our study, the decay of correlation is slower than the exponential decay. Consequently, we describe this phase as _not simply disordered_. While further investigation is required to fully understand this behavior, it likely originates from the unique properties of LLMs, such as infinite-range, non-reciprocal interactions, which go beyond the scope of typical physical systems. In the revised manuscript, we have replaced the expression _not simply disordered_ with a more explicit description of the decay.
>
> # Question 3
> > What are the implications of causality being broken for the conclusions that can be drawn?
>
> By _causality_, we simply mean that an event cannot influence events that occurred in the past. In our study, we interpret $(y_0,\cdots,y_{N-1})$ as a time series. However, in POS tagging, subsequent context (interpreted as _future_ events) can sometimes influence the POS of preceding words (interpreted as _past_ events), thus violating causality. While this may affect our interpretation, the analysis and the conclusion remain valid.  Indeed, we have confirmed similar phase transitions appear when text is mapped to a sequence by replacing each character with a number. In this case, the mapping of each character is independent of the mappings of subsequent characters. We have removed the word _causality_ in the revised manuscript to avoid potential confusion.
>
> If our response does not resolve your concern, it is likely that we have not fully understood your question. Please let us know.

---

> > ### Comment · Reviewer_LsMv · 2024-11-22
> > **Response**
> >
> > Thank you for the clarifications! While the paper is easier to follow and understand, I still feel that the main investigation is rather contrived (it's an external change to the model that doesn’t have any implications about what happens when internal model attributes vary) and does not tell us useful/interesting properties about language models. My score remains unchanged

---

> > > ### Author Response · Authors · 2024-11-28
> > >
> > > We thank the reviewer for the additional feedback. Unfortunately, our previous responses may not have fully addressed your concerns. Although further research is necessary to fully understand the phase transition in LLMs, we believe that our study offers a foundational framework for future investigations that can provide valuable insights.
> > >
> > > To support this, we have included new results in App. E, where we have estimated the transient time and the exponent of its divergence. As discussed in Sec. 5, generated texts are expected to remain natural within this transient time. Therefore, these analyses of the transient time offer quantitative implications on the temperature range where LLMs exhibit behavior close to natural languages.
> > >
> > > Moreover, similar exponents can be defined for parameters other than the temperature parameter. In statistical physics, such exponents are related to each other through a universal relation. Additionally, the set of these exponents is essentially associated with fundamental characteristics of the system. Therefore, applying the statistical analyses presented in our study to other parameters could yield valuable insights into the underlying mechanisms of LLMs.

---

### Author Response · Authors · 2024-11-19

We sincerely thank the reviewers for their careful reading of our manuscript and their valuable feedback. In response to the comments, we have revised our manuscript to address specific concerns. We have removed potentially confusing terminology, added explanations to make it clearer for a broader range of readers, and highlighted the significance of our research. Furthermore, we modified the title to include _temperature-induced_ to clarify our focus. If there are any other concerns, we would be happy to further revise our manuscript.

In the original manuscript, we observed the phase transition in models of sizes 70M, 124M, and 361M. To further support our claims, we have performed additional numerical analyses on Pythia of size 1B, showing that this model also exhibits a similar phase transition (see App. J for the numerical results). These indicate that the phase transition occurs in even larger models.

Finally, we would like to highlight the following key points:
- Based on the formal definition of phase transitions, we present convincing evidence to support the presence of a phase transition.
- We expect that the phase transition can be observed robustly, as discussed in Sec. 6. Indeed, we have observed phase transitions in other models with different mappings, as shown in Apps. I and J.
- Our finding of the phase transition allows us to approach the understanding of LLMs from the physical point of view. For example,
    - The power-law decay in correlation is necessary for texts to be natural. Studying phase transitions and critical phenomena in LLMs would provide a clear understanding of how and what characteristics of natural languages LLMs recognize.
    - Existing statistical-mechanical models do not have phase transitions with the qualitatively same properties observed in this study. This highlights the nature of LLMs and natural languages beyond typical physical systems.
    - The phase transition in every LLM should belong to a _universality class_, as discussed in Sec. 6. Dimensionality and symmetry of the system and the emerging order at the phase transition control its universality class. Thus, identifying the universality classes of LLMs directly classifies the models and should tell us how much a given LLM has mimicked natural language.
- Our analysis can be extended to examine changes induced by not only the temperature parameter but also any other parameters.

In summary, we have proposed a novel framework for characterizing and analyzing the behavior of LLMs. We believe that it will be crucial in advancing future practical developments in LLMs.

---

> ### Author Response · Authors · 2024-11-28
>
> To demonstrate further implications of our research, we have conducted additional analyses of the transient time, within which generated texts are expected to be natural. The results,  presented in App. E, show that the transient time at high temperatures diverges algebraically with an exponent of approximately 1 as $T$ approaches $T_c$. Similar exponents can be defined for parameters other than the temperature parameter. In statistical physics, these exponents, which are related to each other in a universal relation, essentially charatcterize critical phenomena and the fundamental characteristics of the system. Therefore, the estimation of these exponents would provide useful implications for thepretical understanding of LLMs.

---

### Meta-Review · Area_Chair_5ktY · 2025-01-03

**Metareview:**

This paper looks at the range of temperature values at which LLM-generated text mimics the statistical properties of natural language, framing it through the lens of phase transitions in physical systems. The authors hypothesize the existence of a singularity point on the temperature scale where LLM-generated text aligns with natural language properties, with deviations occurring at temperatures above or below this point.

While the perspective is interesting, the experimental setup and evidence supporting the hypothesis are limited. The primary finding is that LLM-generated text preserves the statistical properties of natural text at a temperature of 1. This is not really surprising and also doesn't lead to a deeper understanding of LLMs through the concept of phase transitions. This concern was shared by the reviewers.

The statistical properties examined in the study include long-range correlations of POS tags (specifically, PROPN) and the power spectra of POS tags. However, the rationale behind selecting these properties in drawing parallels with phase transitions is insufficiently justified. The reviewers also raised concerns about older and smaller LLMs being included in the experiments (models: GPT-2, pythia etc., size: <1B), essentially, not providing a complete view of the current state of LLMs. Lastly, the clarity of the writing was a recurring concern among reviewers, making the paper difficult to follow.

Overall, the study is further by its reliance on older and smaller LLMs for experiments, its narrow focus on two statistical properties, and the lack of novelty in its findings. After the author-reviewer discussion, two reviewers expressed weak acceptance (scores of 6 and 6), while the remaining two reviewers recommended rejection (scores of 3 and 3). Based on these considerations, I lean towards rejecting the paper.

**Additional Comments On Reviewer Discussion:**

See above.

---

### Decision · Program_Chairs · 2025-01-22

Reject